# Identifying Coarse-grained Independent Causal Mechanisms with Self-supervision

**Xiaoyang Wang**                                                                    XW28@ILLINOIS.EDU
**Klara Nahrstedt**                                                                  KLARA@ILLINOIS.EDU
**Sanmi Koyejo**                                                                     SANMI@ILLINOIS.EDU
*University of Illinois at Urbana-Champaign*

**Editors:** Bernhard Schölkopf, Caroline Uhler and Kun Zhang

## Abstract

Among the most effective methods for uncovering high dimensional unstructured data's generating mechanisms are techniques based on disentangling and learning independent causal mechanisms. However, to identify the disentangled model, previous methods need additional observable variables or do not provide identifiability results. In contrast, this work aims to design an identifiable generative model that approximates the underlying mechanisms from observational data using only self-supervision. Specifically, the generative model uses a degenerate mixture prior to learn mechanisms that generate or transform data. We outline sufficient conditions for an identifiable generative model up to three types of transformations that preserve a coarse-grained disentanglement. Moreover, we propose a self-supervised training method based on these identifiability conditions. We validate our approach on MNIST, FashionMNIST, and Sprites datasets, showing that the proposed method identifies disentangled models – by visualization and evaluating the downstream predictive model's accuracy under environment shifts.

**Keywords:** Causal mechanisms, generative model, disentanglement, identifiability

## 1. Introduction

The independent causal mechanisms (ICM) principle assumes that a system's variables are generated by autonomous modules (i.e., mechanisms) that do not influence or inform each other (Schölkopf et al., 2012; Peters et al., 2017; Schölkopf, 2019). In this paper, we consider a class of systems where an observable variable follows a mixture distribution, and each mixture component is generated by a composition of mechanisms (Locatello et al., 2018b; Parascandolo et al., 2018). Specifically, each data generating mechanism generates data samples for a specific mixture component, and the data transformation mechanisms could transform the data samples without moving the data samples from one mixture component to another. Moreover, we let each mechanism take latent variables as additional input, which controls the generated data's variation. This kind of data is ubiquitous, and examples include visual data in object recognition tasks, where animals, plants, and other objects are generated by different data generating mechanisms, and the data transforming mechanisms control the background and the lighting condition, among others. Figure 2(*b*) shows an example of our system with three mechanisms that collaboratively generate data samples (i.e., handwritten digits) for a mixture distribution with two mixture components. This manuscript proposes a method

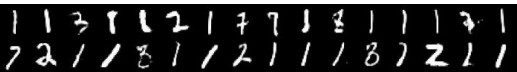

(a) Sampled Data from Mechanisms Learned by Our ICM Model.

(b) Sampled Data from Mechanisms Learned by Existing Competitive Training-based Approach.

Figure 1: Each row represents the data from one mechanism. Ideally, the mechanisms should specialize in synthesizing one digit uniquely. (a) shows that each data generating mechanism specializes in different digits. (b) shows that the mechanisms are confounded with other digits – illustrating a failure of the model and competitive training approach. Figure (b) is adapted from (Locatello et al., 2018b).

for learning the data generating and transforming mechanisms from the observable variable in the aforementioned class of systems by learning the latent variables.

One benefit of learning the data generating and transforming mechanisms is better generalization across environments. Recent works (Koh et al., 2020; Gulrajani and Lopez-Paz, 2021) show that the accuracy of a machine learning (ML) model can decrease drastically if the model is deployed in an environment that differs from the environment where the model is trained. For an image classification task, different environments may have variability in illumination, camera angle, background, color, among others (Koh et al., 2020). As variables like illumination are not conditioned on the data samples or the labels, we can model their effect as data transforming mechanisms controlled by latent variables. Importantly, by the ICM principle, the change of data transformation mechanisms does not influence the data-generating mechanisms. Therefore, if we use the inverted data-generating mechanisms to recover the latent variables that generate the data and feed the latent variables to the downstream predictive ML model, we can expect more robustness to environmental shifts due to disentanglement. Thus, a disentanglement between latent variables results from the disentanglement between mechanisms.

The literature on both independent causal mechanisms (Locatello et al., 2018b; Parascandolo et al., 2018) and disentangled representations (Higgins et al., 2017; Locatello et al., 2020) have produced powerful disentanglement techniques. However, they often use competitive-training-based approaches (Locatello et al., 2018a,b), which are difficult to train, resulting in entanglement between mechanisms. Figure 1(b) illustrates an example where the estimated data-generating mechanisms are entangled – as the mechanisms do not specialize to mixture components in the data. In contrast, using our method, the disentangled mechanisms are specialized to digits 4 and 9 (Figure 1(a)). In recent work, Huang et al. (2019) study clustering datapoints based on shared causal structure but do not incorporate latent variables. Another competitive training method, Parascandolo et al. (2018), learns the mechanisms by letting a mixture of models convert a set of transformed data samples (e.g., rotated data samples) to the original data samples. This approach requires the use of auxiliary data–separate from the original data. However, it is common for the transformed and original data to be mixed in the available dataset.

An additional gap in the literature is that *few of the existing methods for learning the mechanisms analyze identifiability conditions*, i.e., when one can distinguish the desired disentangled model among the entangled models (Locatello et al., 2019; Khemakhem et al., 2020). Existing identifiability results (Shu et al., 2019; Locatello et al., 2020) mainly focus on disentangled repre-

sentations and either need additional labeling efforts or are not compatible with the class of system that we study. Section 3.3 discusses the difficulties of applying previous identifiability results in more detail.

To address the disentanglement, dataset, and identifiability limitations, we seek a practical method that provably identifies the disentangled mechanisms without requiring any auxiliary dataset or label. To this end, we first present a new generative model that uses a degenerate mixture prior distribution to learn the mechanisms. We refer to our new model as the ICM model. Our ICM model first learns the latent variables of the mechanisms and then transforms them into disentangled mechanisms. We define *coarse-grained* disentanglement between latent variables: the variables for the same mechanism could be entangled but each group of variables from different mechanisms should be disentangled. With the proposed ICM model, we prove that we can identify the generative model, which disentangles other latent variables from the latent variables of each data-generating mechanism in a coarse-granularity, up to a limited set of transformations, e.g., transformations that permute the order of the latent variables but do not violate the coarse-grained disentanglement. Albeit coarse, we will show that the benefits of disentanglement under environment shift are preserved. Unlike existing identifiability results (Shu et al., 2019; Locatello et al., 2020; Khemakhem et al., 2020) which require supervision, our identifiability result requires only the invertibility of the generative model. To meet the invertibility condition, we introduce a self-supervised training method, which is easier to implement than competitive training methods (Parascandolo et al., 2018; Locatello et al., 2018b). Further, by enforcing the identifiability conditions, we enforce the coarse-grained disentanglement. Our key contributions are summarized as follows:

- We develop a novel generative model for learning disentangled mechanisms using a degenerate mixture prior.

- We prove the proposed generative model is identifiable up a few transformations that preserve the coarse-grained disentanglement.

- We implement this approach using a self-supervised training method to enforce the identifiability conditions.

We evaluate our approach using synthetic data with known ground-truth latent variables, showing that with an invertible ICM model, our approach closely approximates the true disentangled latent variables, in contrast to non-invertible generative models. Qualitatively, a latent space traversal shows that the latent variables are disentangled at the coarse-granularity of interest. Quantitatively, we show that the proposed ICM model improves downstream classification accuracy under environment shift due to the disentanglement between latent variables. The accuracy improvements are up to 9.75% on MNIST (LeCun et al., 1998), FashionMNIST (Xiao et al., 2017), and Sprites [1] (Li and Mandt, 2018) datasets, compared to VAE (Kingma and Welling, 2013), $\beta$-VAE (Higgins et al., 2017), Ada-GVAE (Locatello et al., 2020), and VaDE (Jiang et al., 2017) models.

## 2. The Generative Model with a Degenerate Mixture Prior

This section introduces our proposed generative model. In constrast to prior work (Parascandolo et al., 2018), our approach includes latent variables for the data generating mechanisms and data

---

1. https://lpc.opengameart.org
   https://github.com/jrconway3/Universal-LPC-spritesheet

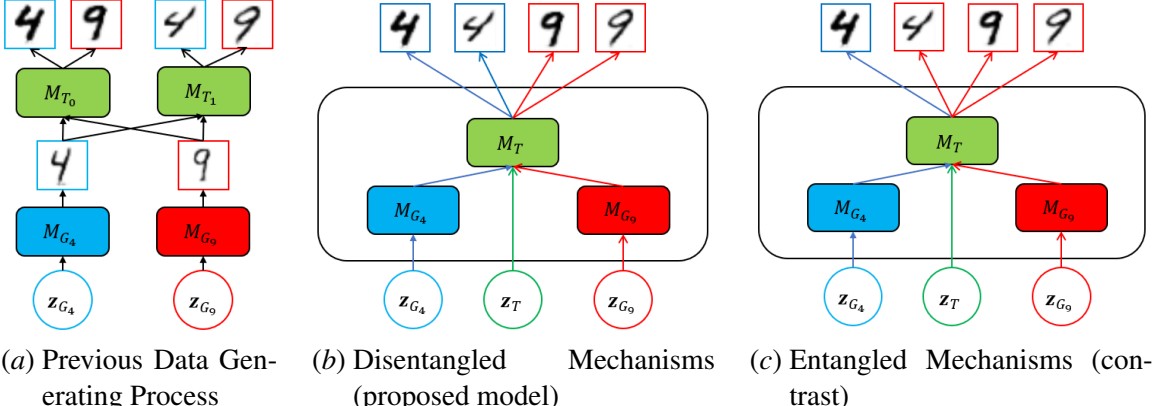

(a) Previous Data Generating Process

(b) Disentangled Mechanisms (proposed model)

(c) Entangled Mechanisms (contrast)

Figure 2: Comparing existing approaches (a) to the proposed approach (b) and an entangled variant (c) without mechanism separation. To see the entanglement, note that the red mechanism for 9s generates the 4. $M_{G_i}$ is a data generating mechanism. $M_{T_i}$ and $M_T$ are data transforming mechanisms. The latent variables and data samples correspond to the mechanisms with the same color. Mechanisms are disentangled if their input latent variables are disentangled and specialized for different types of variation.

transforming mechanisms simultaneously – without requiring an auxiliary dataset. Next, we connect the generative model and latent variables to underlying mechanisms. We further discuss identifiability and propose a training method for our model in Sections 3 and 4, respectively.

### 2.1. Generative Model

**Assumed Data Generating Process** Following prior work (Parascandolo et al., 2018), we assume a two-step deterministic data generating process. A data sample $x' = M_{G_i}(z_{G_i})$ is generated by a data generating mechanism $M_{G_i}$ using latent variables $z_{G_i}$ as in Locatello et al. (2018b). The data sample $x'$ is further transformed to another data sample $x = M_{T_j}(x')$ by a data transformation mechanisms $M_{T_j}$ similar to Parascandolo et al. (2018). Our approach is visualized in Figure 2(a) for MNIST digits 4 and 9. Here, the latent variable $z_{G_4}$ may control whether digit 4 has a bar, crossing its stem while the latent variable $z_{G_9}$ may decide the circle size of digit 9, as the two rows in Figure 1(a) shows. We consider, for example, thickening or rotating a digit as data transforming mechanisms ($M_{T_0}$ and $M_{T_1}$ in Figure 2(a)).

**Our Approach and Model Structure** To learn the mechanisms, we take a two-step approach: (1) designing a model for learning the latent variables and (2) transforming the model to the mechanisms. In step (1), we adopt a model: $x = g(z_G, z_T)$ where $x$ is the data sample, $z_G = [z_{G_0}, z_{G_1}, ...]$ are the latent variables for the data generating mechanisms and $z_T$ is the latent variable for the data transforming mechanisms. By fitting our model to the dataset (with additional techniques), we let each $z_{G_i}$ $z_T$ capture the variability in data. We use a running example to introduce step (2).

**Running Example** Figure 2(b) visualizes a disentangled version of our model. Here, consider a modular model (the black rectangle) that contains the mechanisms as sub-modules. The modular

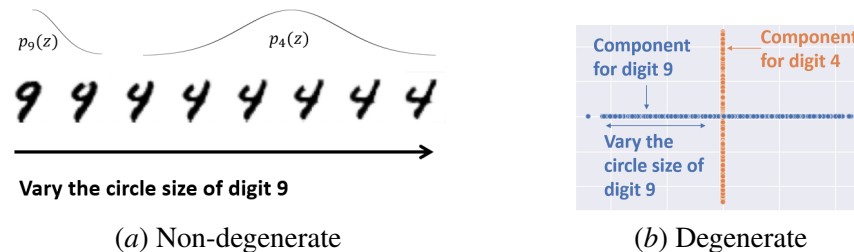

(*a*) Non-degenerate           (*b*) Degenerate

Figure 3: (a) Using non-degenerate mixture prior leads to: (1) One latent variable that encodes multiple types of variations (e.g., for digits 4 and 9). (2) Intervening the latent variable moves the generated data sample from one mixture components to another. (b) Using a degenerate mixture prior avoids these two issues.

model generates data by activating a sequence of sub-modules. We assume the sub-modules are activated upon receiving any non-zero input.

**Previous Models and their Limitations**    Previous competitive training-based approaches (Parascandolo et al., 2018; Locatello et al., 2018b) adopt a model structure that is the same as the assumed data generating process: $\boldsymbol{x} = \sum_{j=1}^{K_T} \pi_{T_j} M_{T_j} \big( \sum_{i=1}^{K_G} \pi_{G_i} M_{G_i}(\boldsymbol{z}_{G_i}) \big)$, where $\boldsymbol{\pi}_T$ and $\boldsymbol{\pi}_G$ are one-hot vectors. Unfortunately, learning mixtures of mechanisms is difficult. Their works assume the canonical distribution (Parascandolo et al., 2018), $\boldsymbol{x}' = \sum_{i=1}^{K_G} \pi_{G_i} M_{G_i}(\boldsymbol{z}_{G_i})$ is available, then learns the two mixtures separately (Parascandolo et al., 2018; Locatello et al., 2018b) but such an assumption does not apply to many conventional datasets.

While the proposed approach is designed to disentangle the latent variables, simply fitting the proposed generative model to the dataset and hoping the $\boldsymbol{z}_{G_i}$ and $\boldsymbol{z}_T$ encodes different types of variations for different mechanisms can fail. The issues are twofold: (1) The commonly used disentangled prior $p(\boldsymbol{z}_G, \boldsymbol{z}_T) = \mathcal{N}(\boldsymbol{z}_G \mid \boldsymbol{0}, \boldsymbol{I}) \times \mathcal{N}(\boldsymbol{z}_T \mid \boldsymbol{0}, \boldsymbol{I})$ would result in samples with multiple $\boldsymbol{z}_{G_i} \neq \boldsymbol{0}$. Such a sample will activate multiple data generating mechanism sub-modules (e.g. $M_{G_4}$ and $M_{G_9}$ in Figure 2(*b*)) in parallel. This is at odds with the independent causal mechanisms concept – where we would like a variable to be generated by one mechanism (potentially with sequential composition) (Parascandolo et al., 2018; Locatello et al., 2018b). (2) It is unclear whether one can disentangle each $\boldsymbol{z}_{G_i}$ and $\boldsymbol{z}_T$ due the well-known identifiability issues (Locatello et al., 2019).

### 2.2. A Degenerate Mixture Prior

In order to address the issue related to the disentangled prior, we propose a degenerate mixture prior. This way, we can make sure that if we sample from one component, the irrelevant dimensions of the samples are $\boldsymbol{0}$ (i.e., deactivated). Prior work (Khemakhem et al., 2020) which uses a non-degenerate mixture prior assumes that each mixture component of the data distribution is generated by a separate component of the mixture prior. However, the non-degenerate mixture prior has two limitations in modeling disentangled mechanisms: (1) a single latent variable must encode multiple types of variation, violating assumptions of the data generating process, and (2) intervening on the latent variable may move the generated data samples from one mixture component to another.

To see the limitations, consider the running example. Here, assume digits 4 and 9 are generated by a one dimensional latent variable $z_G$ which follows $p(z_G) = \pi_4 p_4(z_G) + \pi_9 p_9(z_G)$. The latent variable $z_G$ has to encode two types of variations for the two digits. If we intervene on $z_G$ to vary the circle size of digit 9, we may push $z_G$ into the mixture component where digit 4 lies, as Figure 3(a) shows.

To address the outlined limitations, we use a degenerate mixture prior :

$$p(i) \propto \text{unif}(0, K_G), \quad p(\boldsymbol{z}_T) \propto \mathcal{N}(\boldsymbol{z}_T \mid \boldsymbol{0}, \boldsymbol{I})$$

$$p(\boldsymbol{z}_G \mid i) \propto \mathcal{N}(\boldsymbol{z}_G \mid \boldsymbol{0}, \text{diag}(\boldsymbol{e}_i)), \quad p(\boldsymbol{z}_G) \propto \sum_{i=1}^{K_G} \frac{1}{K_G} \cdot \mathcal{N}(\boldsymbol{z}_G \mid \boldsymbol{0}, \text{diag}(\boldsymbol{e}_i)) \tag{1}$$

where $i$ is the index of data generating mechanisms, $\boldsymbol{e}_i$ is the $i^{\text{th}}$ standard unit vector. We may let each component have a higher-dimensional, non-overlapping support in the implementation to support multiple latent variables for each mechanism. We assume equal prior weights for each mixture component. However, we note that recent work (Ojha et al., 2020) that studies learnable mixture weights in generative models is complementary. Another advantage of the degenerate prior is that all the samples within the same subspace represent the same type of data. In our running example, no matter how we intervene $\boldsymbol{z}_{G_9}$, the model will always generate a digit 9 as is shown is Figure 3(b). To eliminate the ambiguity when $\boldsymbol{z}_G \approx \boldsymbol{0}$, we append a one-hot index $i$ to $\boldsymbol{z}_G$, following Antoran and Miguel (2019).

**Transforming the Model to Independent Causal Mechanisms** It is straightforward to connect our generative model and latent variables to the mechanisms, complementing the intuition that the modular model generates data by activating a sequence of sub-modules. Consider a generative model $g : \mathcal{Z} \to \mathcal{X}$, two data generating mechanisms $M_{G_0} : \mathcal{Z}_{G_0} \to \mathcal{X}_{G_0}$, $M_{G_1} : \mathcal{Z}_{G_1} \to \mathcal{X}_{G_1}$ and one data transforming mechanism $M_T : \mathcal{X}_G \times \mathcal{Z}_T \to \mathcal{X}$ where $\mathcal{Z} = [\mathcal{Z}_{G_0}, \mathcal{Z}_{G_1}, \mathcal{Z}_T]$ and $\mathcal{X}_G = \mathcal{X}_{G_0} \cup \mathcal{X}_{G_1}$. We have $g([\boldsymbol{z}_{G_0}, \boldsymbol{0}, \boldsymbol{z}_T]) = M_T(M_{G_0}(\boldsymbol{z}_{G_0}), \boldsymbol{z}_T)$ and $g([\boldsymbol{0}, \boldsymbol{z}_{G_1}, \boldsymbol{z}_T]) = M_T(M_{G_1}(\boldsymbol{z}_{G_1}), \boldsymbol{z}_T)$. If latent variables $\boldsymbol{z}_{G_0}$, $\boldsymbol{z}_{G_1}$, and $\boldsymbol{z}_T$ are disentangled and encode different type of variations, the corresponding mechanisms $M_{G_0}$, $M_{G_1}$, and $M_T$ are disentangled as a consequence. We will show methods for enforcing the disentanglement in the following sections.

## 3. Identifiability Analysis

Now, we have outlined a compact way to model the mechanisms – but fitting the model to the data is not equivalent to learning the *disentangled* mechanism. Conventional training approaches (Kingma and Welling, 2013; Higgins et al., 2017) optimize the generative model $g : \mathcal{Z} \to \mathcal{X}$ such that $p_g(\boldsymbol{x}) \approx p_{g^*}(\boldsymbol{x})$, where $p_g(\boldsymbol{x})$ is the marginal distribution of model $g$, and $g^*$ is the true disentangled model. However, it is not guaranteed that we have $g = g^*$ even if the loss goes to zero (Locatello et al., 2019). Given the ground-truth disentangled model, there exist infinitely many equivalent models, including entangled models, that achieve the same loss (Locatello et al., 2019). For instance, in our running example, the model could let mechanisms $M_{G_9}$ also generate certain digit 4 when entangled, as Figure 2(c) shows, which violates the disentanglement between mechanisms but does not affect the marginal. Other work (Locatello et al., 2019), which studies generative models with an isotropic Gaussian prior, shows that one can construct infinitely many $g^* \circ h$, where $h : \mathcal{Z} \to \mathcal{Z}$, such that $g^* \circ h$ is completely entangled but $p_{g^* \circ h}(\boldsymbol{x}) = p_{g^*}(\boldsymbol{x})$. Following

prior work (Locatello et al., 2020; Khemakhem et al., 2020), we first formulate the definition of identifiability.

### 3.1. Definition of Identifiability

The idealized goal of ensuring that $p_g(\boldsymbol{x}) = p_{g^*}(\boldsymbol{x}) \Rightarrow g = g^*$ is often infeasible. In practice, we are interested in models that are identifiable up to a class of transformation (Locatello et al., 2020; Khemakhem et al., 2020). Thus, we introduce the following definitions:

**Definition 1** *We define an equivalence relation on $g : \mathcal{Z} \to \mathcal{X}$ as follows:*

$$g \sim_h \tilde{g} \Leftrightarrow \exists h : g = \tilde{g} \circ h \tag{2}$$

*where $h : \mathcal{Z} \to \mathcal{Z}$ is a smooth invertible function.*

**Definition 2** *We say that $g : \mathcal{Z} \to \mathcal{X}$ is identifiable up to $\sim_h$ (or $h$-identifiable) if:*

$$p_g(\boldsymbol{x}) = p_{\tilde{g}}(\boldsymbol{x}) \Rightarrow g \sim_h \tilde{g} \tag{3}$$

Definition 2 clarifies the assumption of a smooth invertible transformation. This assumption is consistent with the prior work (Locatello et al., 2020) and is considered weak. For a generative model, if the mapping between $\mathcal{Z}$ and $\mathcal{X}$ is injective, that means the generative model fails to induce part of the data distribution. On the other hand, if the mapping is surjective, that means that if we vary $\boldsymbol{z}$, the generated data $\boldsymbol{x}$ may not change. In both cases, the non-bijective generative model is called collapsed model and is unfavorable (Arjovsky et al., 2017; Yang et al., 2019). To make this assumption more realistic, we will show how to enforce the invertibility in Section 4.

### 3.2. Identifiability of ICM Model

With Definitions 1 and 2, we use the following lemma to simplify the discussion in our main result in Theorem 4.

**Lemma 3** *Let $\mathcal{G}$ be the space of smooth invertible functions with smooth inverse (i.e., a diffeomorphism) (Locatello et al., 2020) that map $\mathcal{Z}$ to $\mathcal{X}$, and $h : \mathcal{Z} \to \mathcal{Z}$ is a smooth invertible function. Then, any function $g \in \mathcal{G}$ can be represented as $g = g^* \circ h$, where $g^* : \mathcal{Z} \to \mathcal{X}$ is the assumed true disentangled generative model in the function space $\mathcal{G}$. Formally, we have $g \sim_h g^*, \forall g \in \mathcal{G}$ and the model $g$ is $h$-identifiable.*

We can prove this lemma by simply letting $h = (g^*)^{-1} \circ g$. A detailed proof is in Appendix A.1. This lemma guarantees that by optimizing towards $p_g(\boldsymbol{x}) = p_{g^*}(\boldsymbol{x})$, the learned model $g$ equals to the true $g^*$ up to a smooth invertible transformation $h$.

Now, we present our ***main identifiability result***. We show that the function $h$, when paired with our degenerate prior, does not entangle the latent variables of other mechanism to those of any data-generating mechanisms. The latent variables are disentangled in the sense that $h$ has to map each latent variable $\boldsymbol{z}_{G_i}$ to disjoint subspaces. For example, the function $h$ can not map $\boldsymbol{z}_{G_9}$ to the subspace where $\boldsymbol{z}_{G_4}$ lies in. As such, in our running example, the model can not generate a digit 4 by manipulating $\boldsymbol{z}_{G_9}$. Also, the function $h$ can not map $\boldsymbol{z}_T$ to any $\boldsymbol{z}_{G_i}$, which preserves the benefit of better generalization across environments.

As all the smooth invertible generative models are $h$-identifiable, we only need to consider how the function $h$ entangles the latent variables $\boldsymbol{z}$ before the true generative model. Note that we do not assume $h$ is always $(g^*)^{-1} \circ g$. The main idea is: with a degenerate mixture prior, if $h$ maps one latent variable to the two or more different subspaces, then the result has measure zero, or, if we transform back to the original space using $h^{-1}$, the result has measure zero, which violates the definition of $h$.

**Theorem 4** *Let $i \in \{1, 2, ..., K_G\}$ be the index of data generating mechanisms, let $\sigma : \mathbb{N}^+ \to \mathbb{N}^+$ be a bijective function that permutes the index $i$, let $\boldsymbol{z}$ be a latent variable that follows the degenerate mixture distribution defined in Eq. (1), and let $\mathcal{Z}_{G_i}$, $\mathcal{Z}_T$, and $\mathcal{Z}$ be the support of $\boldsymbol{z}_{G_i}$, $\boldsymbol{z}_T$ and $\boldsymbol{z}$, respectively. We let $\hat{\boldsymbol{z}} = h(\boldsymbol{z})$, $\hat{\mathcal{Z}}$ be the support of $\hat{\boldsymbol{z}}$. We assume $g^* : \mathcal{Z} \to \mathcal{X}$ is the true disentangled model. Then, if there exists a smooth invertible function $h : \mathcal{Z} \to \mathcal{Z}$ such that $g \sim_h g^*$, then $h$ maps each $\mathcal{Z}_{G_i}$ to $\hat{\mathcal{Z}}_{G_{\sigma(i)}}$, disjoint from $\hat{\mathcal{Z}}_{G_{\sigma(j)}}, \forall j \neq i$, as well as $\hat{\mathcal{Z}}_T$, and maps $\mathcal{Z}_T$ to $\hat{\mathcal{Z}}_T$, which is disjoint from $\hat{\mathcal{Z}}_{G_{\sigma(i)}}, \forall i \in \{1, ..., K_G\}$.*

The proof of Theorem 4 is in Appendix A.2. We now discuss the remaining three types of transformations of $h$. First, we cannot (and do not need to) guarantee the $i^{\text{th}}$ learned subspace $\hat{\mathcal{Z}}_{G_i}$ corresponds to the $i^{\text{th}}$ true subspace $\mathcal{Z}_{G_i}$ because the disentanglement between latent variables will not be affected. Therefore, the learned latent variables $\hat{\boldsymbol{z}}_G$ may be subspace-wise permutation of the true latent variables $\boldsymbol{z}_G$. Second, the function $h$ may still entangle the latent variables of the same mechanism, which is the case for an intra-subspace smooth invertible transformation. Third, $h$ may entangle $\boldsymbol{z}_G$ to $\boldsymbol{z}_T$ in $\hat{\boldsymbol{z}}_T$, which does not affect the disentanglement of $\hat{\boldsymbol{z}}_G$. Although the disentanglement has a coarse granularity, we will empirically show that we can still benefit from the coarse-grained disentanglement.

### 3.3. Comparison to Other Identifiability Results

**Identifiable VAE** Previous work (Khemakhem et al., 2020) combines non-linear independent component analysis (ICA) and variational autoencoder (VAE) for identifiability by leveraging the invertibility of the sufficient statistics parameter matrix. The latent variables are conditioned on a label in their setting: the class label, the environment label, or others. Besides the label requirement, the invertibility prevents the mechanisms from sharing or duplicating the latent variables because the matrix, of which each column is the parameters for a label, will be low-rank. Consequently, this method is not applicable when there are additional data transforming mechanisms.

**Weakly-supervised Disentanglement** Ada-GVAE (Locatello et al., 2020) achieves identifiability by introducing pair observations that share the same value on at least one latent variable. However, even in a simple dataset like MNIST, this assumption may not hold because two digits may differ w.r.t. each latent variable such as stroke-thickness, width, or brightness. Other work (Shu et al., 2019) adopts a weaker assumption by introducing rank pairing. However, this method requires additional supervision and certain latent variables (e.g., style) are hard to compare or rank.

## 4. Proposed Model Architecture and Self-supervised Training Method

In practice, we need to enforce the generative model $g$ to be smooth invertible with smooth inverse. As the neural network is commonly assumed as smooth (Xie et al., 2020; Liang et al., 2021), this section shows how to enforce the invertibility assumption. For an invertible generative model, we have $z = g^{-1}(g(z))$. If we can find $g^{-1}$ such that $z = g^{-1}(g(z))$ holds, then the generative model $g$ is invertible.

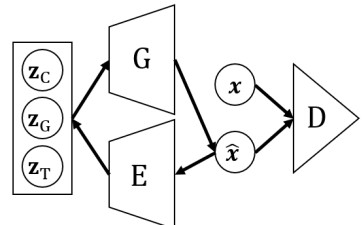

Figure 4: Model Architecture

To do so, we first design a new model architecture. Technically, we compose a generator network $G : \mathcal{Z} \to \mathcal{X}$, a discriminator network $D : \mathcal{X} \to \mathcal{Y}$, and an auxiliary encoder network $E : \mathcal{X} \to \mathcal{Z}$. We append the encoder to the generator's output and connect the encoder back to the generator's input as Figure 4 shows. Specifically, our generator network and discriminator networks are from the Wasserstein GAN (Arjovsky et al., 2017; Gulrajani et al., 2017). Our auxiliary encoder is a deterministic encoder that approximates $g^{-1}$.

We use self-supervised training to jointly train the generator and the encoder. The self-supervision involves predicting the latent variables that generate the data samples using the auxiliary encoder network $E$. The loss function is:

$$\mathcal{L} = W(\mathbb{P}_g || \mathbb{P}_r) + \lambda_G \mathbb{E}_{z \sim p(z)} ||z_G - E(G(z))_G||_2^2 + \lambda_T \mathbb{E}_{z \sim p(z)} ||z_T - E(G(z))_T||_2^2$$
$$+ \lambda_C \mathbb{E}_{z \sim P(z)} \mathcal{H}(z_C, E(G(z))_C) + \lambda_R \mathbb{E}_{x \sim p(x)} ||x - G(E(x))||_2^2 \tag{4}$$

where $\mathbb{P}_g$ denotes the distribution induced by the generator network $G$, $\mathbb{P}_r$ denotes the real data distribution, $W(\mathbb{P}_g || \mathbb{P}_r)$ denotes the Wasserstein distance (Arjovsky et al., 2017) between two distributions, $z = [z_C, z_G, z_T]$, $z_C$ is a categorical variable representing the index of the data generating mechanisms and its value is determined by $z_G$, $z_G$ and $z_T$ are the same as is defined in Section 2, and $\mathcal{H}$ denotes the cross-entropy loss.

The loss function above can be decomposed into four parts: 1) $W(\mathbb{P}_g || \mathbb{P}_r)$ is the WGAN loss (Gulrajani et al., 2017). 2) $\mathbb{E}_{z \sim p(z)} ||z_G - E(G(z))_G||_2^2 + \mathbb{E}_{z \sim p(z)} ||z_T - E(G(z))_T||_2^2$ is the self-supervision loss for invertability. 3) $\mathbb{E}_{z \sim P(z)} \mathcal{H}(z_C, E(G(z))_C)$ is the auxiliary self-supervision loss that encourages the data generated by different mechanisms to be different and separable. 4) $\mathbb{E}_{x \sim p(x)} ||x - G(E(x))||_2^2$ is the cycle-consistent loss, so $x \approx G(E(x))$. The encoder takes data from the generator as input. However, the generated data distribution may differ from the real data distribution. Thus, certain data in the real distribution may become out-of-distribution (OOD) for the encoder. Such distribution divergence hurts the performance of downstream tasks. The cycle consistency mitigates the distribution divergence by encouraging the ICM model to capture the whole real data distribution. While the usage of cycle-consistency has a long history in image-to-image translation (Zhu et al., 2017), visual-tracking (Sundaram et al., 2010), and other areas, the loss in our method is developed from a different motivation and serves a different purpose.

**Model Architecture Comparison**    Similar model architectures have been explored by previous works (Larsen et al., 2016; Bao et al., 2017), which append a discriminator to a VAE model. The issue of the VAE-based model is that the inferred prior could drift away from the true prior (Kumar et al., 2017). Such drift breaks our identifiability result, where the prior has a specific form. Prior work (Kumar et al., 2017) adds the distance between the inferred prior and the true prior as a regularizer to the loss function. However, the regularizer only approximates the true distance because

the exact computation is intractable. As such, it is unclear whether the true distance decreases to zero. In contrast, our design allows the generator to directly sample from the true prior distribution.

## 5. Related Work

**Independent Component Analysis (ICA)** Khemakhem et al. (2020) bridge the gap between the non-linear ICA and generative models. The non-linear ICA-based approach can identify the true distribution over observed and latent variables up to a linear transformation. However, the strong theoretical guarantee comes with strong assumptions as is discussed in Section 3.3.

**Disentangled Representations** Disentangled representations (Higgins et al., 2017; Shu et al., 2019; Locatello et al., 2020) assume that the data is generated by a single data generating mechanism using a set of independent latent explanatory factors (Bengio et al., 2013). In terms of modeling, we assume each mixture component in the data distribution could have their private factors and all the mixture component share a set of shared factors. In terms of disentanglement, we only require the latent explanatory factors associated with different mechanisms to be disentangled, while the factors associated with the same mechanisms could be entangled. Although our disentanglement has a coarse granularity compared to previous results, our disentangled model is provably identifiable without manual labeling, whereas their results need at least weak-supervision. We will also show in the experiment that our coarse-grained identifiability is useful.

## 6. Experiments

This section first shows a visualization of the identifiability on synthetic data where the ground-truth latent variable is available. Then, we evaluate our method both qualitatively and quantitatively. Qualitatively, we conduct latent space traversal. Quantitatively, we develop a method to evaluate the disentanglement by measuring the accuracy of downstream classifiers under environment shifts, similar to existing works (Locatello et al., 2020). Additional ablation studies in Appendix C show the importance of the loss terms in Eq. (4) and the benefit of using a degenerate mixture prior. The downstream classifiers predict the label of data samples. If the latent variables are better disentangled, the environment shift impacts the downstream classifier less. We briefly introduce the baselines and data setup and detail the setting in Appendix B.

**Baseline Setup** Existing competitive training-based methods (Parascandolo et al., 2018) for learning independent causal mechanisms do not apply to our dataset because the auxiliary dataset is unavailable, as discussed in Section 1. For the other competitive training-based methods (Locatello et al., 2018b), we have seen that the mechanisms are easily entangled (as shown in Figure 1(*b*)). Hence, we select models that focus on disentangled representations, VAE, $\beta$-VAE, and AdaGVAE, as our baselines. We also compare our method to a clustering approach VaDE.

**Data Setup** We generate synthetic data by changing the height, width and brightness of digits from MNIST dataset. For the quantitative evaluation, we use the encoder network $E$ to predict latent variables and feed to prediction to downstream ML models. For experiments, we use 1) three datasets, which are MNIST (MN), FashionMNIST (FMN), and Sprites (SP). Sprites dataset contains animated characters. We select character action and color as the types of variations. 2) five environment shifts, all of which already exist in the data.

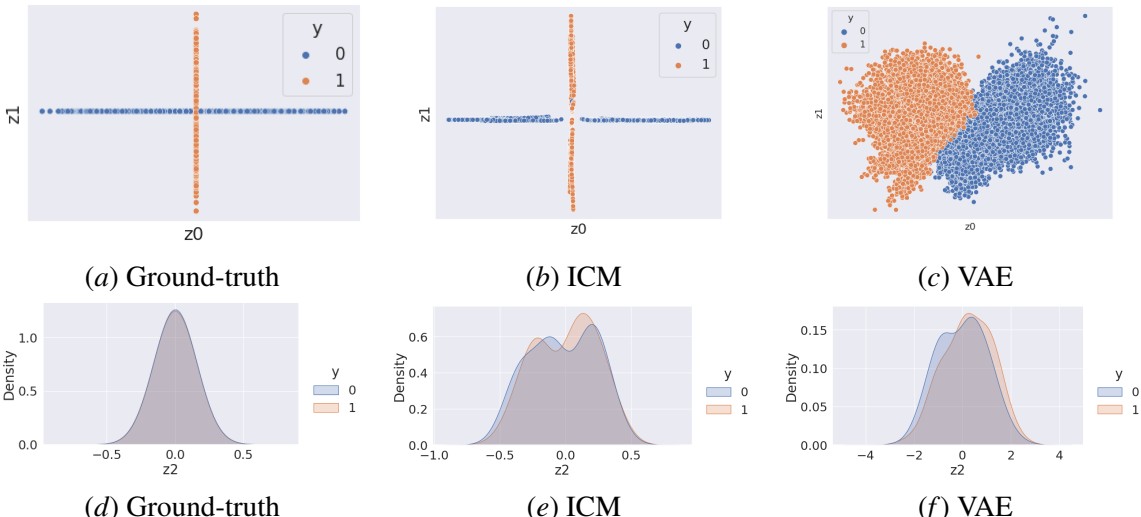

Figure 5: The true distribution and the estimated distribution from our ICM model and VAE. (a)-(c) plot $z_0$ and $z_1$ that should jointly follow a degenerate mixture distribution, (d)-(e) plot $z_2$ that should follow a Gaussian distribution. $y$ is the label. Our ICM model closely approximates the true distribution while VAE fails to recover $z_0$ and $z_1$.

## 6.1. Simulation on Synthetic Data

We define three latent variables: $z_0$ (height), $z_1$ (width), and $z_2$ (brightness). The variables $z_0$ and $z_1$ jointly follow a degenerated mixture distribution, and $z_3$ follows a conventional Gaussian distribution. We then sample two images from the MNIST dataset, which are labeled as $y = 0$ and $y = 1$, respectively. The synthetic dataset is generated by varying the height of the first image, the width of the second image, and the brightness of both images. We fit our ICM model, with enforcing invertibility, and a VAE model, without enforcing invertibility, to the synthetic dataset. Figure 5 shows the estimated distributions of the degenerated mixture part and the Gaussian part. Our ICM model closely approximates the true distribution while the VAE model fails to preserve the degeneracy of the distribution.

## 6.2. Qualitative Evaluation

We show the latent space traversal of ICM model on MNIST dataset in Figure 6. Specifically, we set the $z_C$ to the correct mechanism for each row and set and the latent variables that are not being traversed to 0. The visualization result illustrates the disentanglement as the traversal of each subspace shows a different type of variation. Furthermore, we show that each data generating mechanism's traversal yields a distinct, mechanism-specific type of variation. The traversal of the data transforming mechanism produces the same type of variation. We show the traversal of $\beta$-VAE in Figure 7 as a comparison. Due to the limited space, we report more visualization on MNIST, FashionMNIST and Sprites dataset in Appendix C.4. The visualization also shows that our ICM model is robust to interventions (Suter et al., 2019) in the sense that no matter how we change $z_{G_i}$

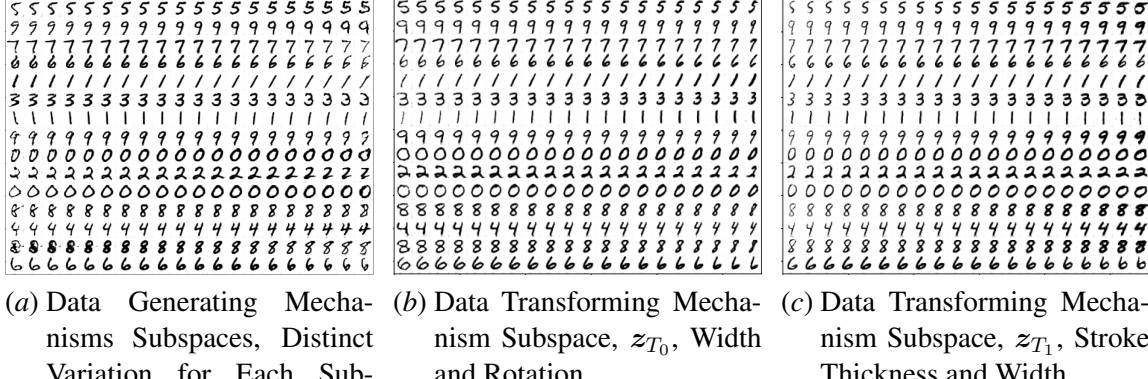

(a) Data Generating Mecha-
nisms Subspaces, Distinct
Variation for Each Sub-
space/Row

(b) Data Transforming Mecha-
nism Subspace, $z_{T_0}$, Width
and Rotation

(c) Data Transforming Mecha-
nism Subspace, $z_{T_1}$, Stroke
Thickness and Width

Figure 6: Latent Space Traversal of ICM Model on MNIST

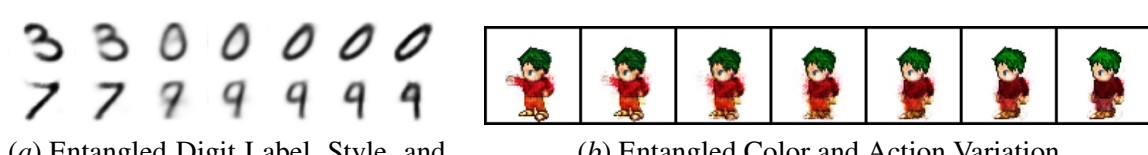

(a) Entangled Digit Label, Style, and
Azimuth Variation

(b) Entangled Color and Action Variation

Figure 7: Latent Space Traversal of $\beta$-VAE Model on MNIST and Sprites

and $z_T$ while keeping $z_C$ and $z_{\setminus G_i}$ fixed, the generator does not generate data that does not belong to mechanism $G_i$ (e.g., digit 4 does not change to digit 9 during the traversal).

We note that the number of data generating mechanisms is chosen to be larger than the number of classes (e.g. 15 mechanisms for 10 classes in MNIST), which means one true mechanism could be split into two estimated mechanisms. This setting is consistent with previous works (Parascandolo et al., 2018; Locatello et al., 2018b). Appendix B proposes a procedure for choosing the number of mechanisms.

### 6.3. Quantitative Evaluation

To quantitatively measure the accuracy of downstream ML models under environment shift, we first partition the dataset using the learned latent variables, from a pre-trained ICM model, as the conditions that are shifted by the environment. The shifted conditions include stroke thickness (T), width and rotation (W&R), rotation (R), darkness and width (D&W), and color. Figure 14 in the Appendix further visualizes these conditions. Since the true $z_T$ is uniquely determined by $\hat{z}_T$ and the true $z_G$, for any data sample $x = g(h(z))$ with a given $z_G$, varying is equivalent to varying the true even if $\hat{z}_T$ entangles other variables. Then, we partition the MNIST and FashionMNIST dataset into subsets $\{x \mid x \in \mathcal{X}, \hat{z}_i = E(x)_i \in [C_{lower}, C_{upper})\}$ using the predicted value of the condition, which is in the $i^{th}$ dimension if $\hat{z}$. For the train set, we set $[C_{lower}, C_{upper})$ to $[0, \infty)$. For each test set, we set $[C_{lower}, C_{upper})$ to $[-0.1, 0), [-0.2, -0.1), ..., [-3.0, -2.9)$. We use the $C_{lower}^{train} - C_{upper}^{test}$ to represent the *shift distance*. For the Sprites dataset, the train set has randomly colored images and the test set has images with only red and green components in the RGB color

space, similar to prior works that also consider color shift (Locatello et al., 2020; Dittadi et al., 2021).

Next, we train the ICM model and other competitors on the MNIST and FashionMNIST train set and Sprites train and test set. Appendix B discusses the training setting on Sprites. Then, we use gradient boosted decision trees (GBT) from the Scikit-learn library (Pedregosa et al., 2011) with default parameters as the downstream ML model, which is the same as previous works (Locatello et al., 2019, 2020). We train the GBT with 1000 train samples (MN, FMN) or 50 train samples (Sprites), which is sufficient for GBT to produce decent accuracy, and test the GBT accuracy on a sequence of test sets. The test repeats ten times on each test set. Table 1 shows the average accuracy and accuracy variations. Our ICM model *achieves the best average accuracy* across all the experiments except for MNIST (R). Due to the limited space, we analyze the experiments on the MNIST (R) dataset in the Appendix C.5.

Table 1: Average Accuracy of Downstream Classifiers under Different Shift Strength

| MODEL | MN (T) | MN (W&R) | MN (R) | FMN (D&W) | SPRITES | AVERAGE |
|---|---|---|---|---|---|---|
| ICM | **74.63%** $_{2.43\%}$ | **56.91%** $_{3.70\%}$ | 43.71% $_{7.10\%}$ | **46.61%** $_{5.24\%}$ | **99.25%** $_{1.79\%}$ | **64.22%** |
| VAE | 55.91% $_{4.32\%}$ | 41.92% $_{6.30\%}$ | 39.90% $_{7.33\%}$ | 37.08% $_{9.02\%}$ | 92.99% $_{4.89\%}$ | 53.56% |
| $\beta$-VAE | 73.34% $_{2.71\%}$ | 46.33% $_{4.98\%}$ | 46.05% $_{7.80\%}$ | 41.60% $_{7.83\%}$ | 98.75% $_{2.68\%}$ | 61.21% |
| ADA | 68.09% $_{2.96\%}$ | 47.16% $_{4.90\%}$ | 47.58% $_{7.31\%}$ | 40.85% $_{7.98\%}$ | 98.75% $_{3.11\%}$ | 60.49% |
| VADE | 73.64% $_{2.69\%}$ | 48.57% $_{4.92\%}$ | **48.54%** $_{7.16\%}$ | 43.21% $_{7.35\%}$ | 98.25% $_{2.95\%}$ | 62.44% |

[*] Letters in parenthesis are abbreviations of the conditions shift across environments.
The small numbers represent accuracy variations.

## 7. Conclusion and Future Work

This paper presents a generative model with a degenerate prior, called the ICM model, to learn a broad range of practical mechanisms that generate and transform data. We analyze conditions under which the disentangled ICM model is identifiable. To enforce the identifiability conditions, we develop a model architecture and a self-supervised training method for the ICM model. Empirical results show that the ICM model achieves better disentanglement and improves the accuracy of downstream prediction ML models under environment shifts.

A straightforward extension of this work is to consider a class of systems where the observational data is generated by a sequential composition of data generating mechanisms, or the data generating mechanisms may share some latent variables.

## Acknowledgments

Koyejo acknowledges partial funding from a Sloan Fellowship and a Strategic Research Initiatives award from the University of Illinois at Urbana-Champaign. This work was also funded in part by NSF 2046795, 1909577, 1934986, 1922758, 1827126, 1835834, 2126246, and NIFA award 2020-67021-32799. The authors are very grateful to Dr. Hongbin Pei for proofreading multiple iterations of this manuscript and the anonymous reviewers for the suggestions on improving the presentation of this work.

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

## Appendix A. Proofs

### A.1. Proof of Lemma 3

**Lemma 3** *Let $\mathcal{G}$ be the space of smooth invertible functions with smooth inverse (i.e., a diffeomorphism) (Locatello et al., 2020) that map $\mathcal{Z}$ to $\mathcal{X}$, and $h : \mathcal{Z} \to \mathcal{Z}$ is a smooth invertible function. Then, any function $g \in \mathcal{G}$ can be represented as $g = g^* \circ h$, where $g^* : \mathcal{Z} \to \mathcal{X}$ is the assumed true disentangled generative model in the function space $\mathcal{G}$. Formally, we have $g \sim_h g^*, \forall g \in \mathcal{G}$ and the model $g$ is $h$-identifiable.*

**Proof** By definition, $g : \mathcal{Z} \to \mathcal{X}$ and $g^* \circ h : \mathcal{Z} \to \mathcal{X}$ are equal if their domain and codomain are the same and $g(\boldsymbol{z}) = g^*(h(\boldsymbol{z}))$ for any $\boldsymbol{z} \in \mathcal{Z}$. For $g \in \mathcal{G}$ and $g^* \circ h$, their domain and codomain are defined to be the same. Then, we could let $h$ to be a permutation function, which is invertible, that satisfies $g(\boldsymbol{z}) = g^*(h(\boldsymbol{z})), \forall \boldsymbol{z} \in \mathcal{Z}$. First, we choose a $\boldsymbol{z} \in \mathcal{Z}$ and let $\boldsymbol{x} = g(\boldsymbol{z})$. Then, we can construct $\hat{\boldsymbol{z}} = h(\boldsymbol{z})$ such that $g^*(\hat{\boldsymbol{z}}) = g^*(h(\boldsymbol{z})) = \boldsymbol{x}$. We can apply this procedure to every $\boldsymbol{z} \in \mathcal{Z}$. This is because we assume the mapping between $\mathcal{Z}$ and $\mathcal{X}$ are bijective, which means there is no conflict. As we assume $h$ to be smooth, which is not implicitly satisfied by the permutation function, we show the smooth assumption holds by showing the limit of $h$ exists everywhere on $\mathcal{Z}$:

$$
\begin{aligned}
&\lim_{\Delta\boldsymbol{z}\to\boldsymbol{0}} \frac{h(\boldsymbol{z}+\Delta\boldsymbol{z}) - h(\boldsymbol{z})}{\Delta\boldsymbol{z}} \\
&= \lim_{\Delta\boldsymbol{z}\to\boldsymbol{0}} \frac{g^{*^{-1}}(g(\boldsymbol{z}+\Delta\boldsymbol{z})) - g^{*^{-1}}(g(\boldsymbol{z}))}{\Delta\boldsymbol{z}} \\
&= g^{*^{-1}{}'}(g(\boldsymbol{z})) \times g'(\boldsymbol{z})
\end{aligned}
\tag{5}
$$

As we know that both $g$ and $g^{*^{-1}}$ are smooth, their composition, $g^{*^{-1}} \circ g$, are also smooth. Thus, the limit above exist for any $\boldsymbol{z} \in \mathcal{Z}$. By definition, $h$ is a smooth function. Thus, we can always find a smooth invertible permutation function $h$ for any $g \in \mathcal{G}$ such that $g = g^* \circ h, \forall g \in \mathcal{G}$. As the valid output $\boldsymbol{x}$ for $g$ and $g^* \circ h$ identical, we have $g \sim_h g^*$ by definition. As the equivalence relation holds between any $g \in \mathcal{G}$ and $g^*$ that admits the same marginal distribution, $g$ is $h$-identifiable. ∎

### A.2. Proof of Theorem 4

**Theorem 4** *Let $i \in \{1, 2, ..., K_G\}$ be the index of data generating mechanisms, let $\sigma : \mathbb{N}^+ \to \mathbb{N}^+$ be a bijective function that permutes the index $i$, let $\boldsymbol{z}$ be a latent variable that follows the degenerate mixture distribution defined in Eq. (1), and let $\mathcal{Z}_{G_i}$, $\mathcal{Z}_T$, and $\mathcal{Z}$ be the support of $\boldsymbol{z}_{G_i}$, $\boldsymbol{z}_T$ and $\boldsymbol{z}$, respectively. We let $\hat{\boldsymbol{z}} = h(\boldsymbol{z})$, $\hat{\mathcal{Z}}$ be the support of $\hat{\boldsymbol{z}}$. We assume $g^* : \mathcal{Z} \to \mathcal{X}$ be the true disentangled model. Then, if there exists a smooth invertible function $h : \mathcal{Z} \to \mathcal{Z}$ such that $g \sim_h g^*$, then $h$ maps each $\mathcal{Z}_{G_i}$ to $\hat{\mathcal{Z}}_{G_{\sigma(i)}}$, disjoint from $\hat{\mathcal{Z}}_{G_{\sigma(j)}}, \forall j \neq i$, as well as $\hat{\mathcal{Z}}_T$, and maps $\mathcal{Z}_T$ to $\hat{\mathcal{Z}}_T$, which is disjoint from $\hat{\mathcal{Z}}_{G_{\sigma(i)}}, \forall i \in \{1, ..., K_G\}$.*

**Proof** By the definition of degenerate mixture prior, we have: if $\boldsymbol{z}_{G_i} \neq \boldsymbol{0}$, $\boldsymbol{z}_{G_j} = \boldsymbol{0}$ for all $j \neq i$. We name this condition as the structure constraint. As we know that $h : \mathcal{Z} \to \mathcal{Z}$ and $g^* : \mathcal{Z} \to \mathcal{X}$, we have $\mathcal{Z} = \hat{\mathcal{Z}}$ because the valid inputs for $g^*$ are fixed. The structure constraint is enforced on both $\mathcal{Z}$ and $\hat{\mathcal{Z}}$. If $h$ entangles $\hat{\boldsymbol{z}}_{G_{\sigma(i)}}$ and $\hat{\boldsymbol{z}}_{G_{\sigma(j)}}$ for any $i \neq j$, which means the change of one ground-truth latent variable $\boldsymbol{z}_{G_k}$ affects two learned latent variables $\hat{\boldsymbol{z}}_{G_{\sigma(i)}}$ and $\hat{\boldsymbol{z}}_{G_{\sigma(j)}}$. Then, there

exist a $z$ such that $h(z)_{G_i} = \hat{z}_{G_{\sigma(i)}} \neq \mathbf{0}$ and $h(z)_{G_j} = \hat{z}_{G_{\sigma(j)}} \neq \mathbf{0}$ hold simultaneously, violating the structural constraint.

To see the violation, we can let $[z_{G_i}, z_{G_j}] = [[1.0, 0.0]^\top, [0.0, 0.0]^\top]$ and construct $[\hat{z}_{G_{\sigma(i)}}, \hat{z}_{G_{\sigma(j)}}] = h([z_{G_i}, z_{G_j}]) = [[0.7, 0.0]^\top, [0.5, 0.0]^\top]$. We can also let $[z_{G_i}, z_{G_j}] = [[1.0, 1.0]^\top, [0.0, 0.0]^\top]$ and construct $[\hat{z}_{G_{\sigma(i)}}, \hat{z}_{G_{\sigma(j)}}] = h([z_{G_i}, z_{G_j}]) = [[0.0, 1.0]^\top, [1.0, 0.0]^\top]$. The existence of such $\hat{z}_{G_{\sigma(i)}}$ and $\hat{z}_{G_{\sigma(j)}}$ violates the structure constraint. Thus, $h$ need to map each $\mathcal{Z}_{G_i}$ to $\hat{\mathcal{Z}}_{G_{\sigma(i)}}$, disjoint from $\hat{\mathcal{Z}}_{G_{\sigma(j)}}, \forall i \neq j$, to make the structure constraint hold.

Additionally, if $h$ entangles $\hat{z}_T$ and $\hat{z}_{G_{\sigma(i)}}$ for any valid $i$, there exist two cases: 1) $h$ maps $\mathcal{Z}_T$ to $\hat{\mathcal{Z}}_T$ and $\hat{z}_{G_{\sigma(i)}}$ or 2) $h$ maps $\mathcal{Z}_{G_i}$ to $\hat{\mathcal{Z}}_T$ and $\hat{z}_{G_{\sigma(i)}}$. For the first case, there exist a $j \neq i$ such that $z_{G_i} = \mathbf{0}$ and $z_{G_j} \neq \mathbf{0}$ but $h(z)_{G_i} = \hat{z}_{G_{\sigma(i)}} \neq \mathbf{0}$ and $h(z)_{G_j} = \hat{z}_{G_{\sigma(j)}} \neq \mathbf{0}$. This is because $z_T$ is independent from all the $z_{G_i}$, $z_{G_j}$ and can push $\hat{z}_{G_{\sigma(i)}}$ to arbitrary value when other variables are fixed. Thus, the structure constraint is violated. The second case falls within our claim.

Furthermore, if $h$ entangles multiple mechanisms, we can find two variables among the entangled latent variables to show that the structure constraint is violated. ∎

## Appendix B. Implementation Details

### B.1. Network Architecture

We report the network architectures for 1x28x28, and 3x64x64 images in Table 2, and 3, respectively. We use the same generator/decoder and encoder (with modified output layer) for VAEs. For the colored dataset Sprites, we use a separate encoder, which takes the gray-scale image is input and predicts $\hat{z}_C$ and $\hat{z}_G$. Such configuration would avoid identifying mechanisms by color.

Table 2: Generator, Discriminator, and Encoder Architectures for $1 \times 28 \times 28$ Inputs

| Generator | Discriminator | Encoder |
|---|---|---|
| Input: $\mathbb{R}^{\dim(z)}$ | Input: $\mathbb{R}^{1 \times 28 \times 28}$ | Input: $\mathbb{R}^{1 \times 28 \times 28}$ |
| FC 1024 BN ReLU | 4×4 conv, 64 LReLU, stride 2 | 4×4 conv, 64 LReLU, stride 2 |
| FC 128×7×7 BN ReLU | 4×4 conv, 128 LReLU, stride 2 | 4×4 conv, 128 LReLU, stride 2 |
| 4×4 upconv, stride 2, 64 BN ReLU | FC 1024 LReLU | FC 1024 LReLU |
| 4×4 upconv, stride 2, 1 Sigmoid | FC 1 | FC $\dim(z)$ |

Table 3: Generator, Discriminator, and Encoder Architectures for $3 \times 64 \times 64$ Inputs

| Generator | Discriminator | Encoder |
|---|---|---|
| Input: $\mathbb{R}^{\dim(z)}$ | Input: $\mathbb{R}^{3 \times 64 \times 64}$ | Input: $\mathbb{R}^{3 \times 64 \times 64}$ |
| FC 1024 BN ReLU | 4×4 conv, 64 LReLU, stride 2 | 4×4 conv, 64 LReLU, stride 2 |
| FC 128×8×8 BN ReLU | 4×4 conv, 64 LReLU, stride 2 | 4×4 conv, 64 LReLU, stride 2 |
| 4×4 upconv, stride 2, 64 BN ReLU | 4×4 conv, 128 LReLU, stride 2 | 4×4 conv, 128 LReLU, stride 2 |
| 4×4 upconv, stride 2, 64 BN ReLU | FC 1024 LReLU | FC 1024 LReLU |
| 4×4 upconv, stride 2, 3 Sigmoid | FC 1 | FC $\dim(z)$ |

### B.2. Hyper-parameters

We set $[\lambda_G, \lambda_T, \lambda_C \lambda_R]$ to $[10.0, 10.0, 30.0, 0.05]$ on all the datasets and change $\lambda_T$ to 50 for Sprites dataset. We train our ICM model for 200 epochs. For each $\beta$-VAE and Ada-GVAE model, we train the model for 200 epochs with $\beta = [1, 2, 4, 8, 16]$. We train each generative model three times and report the best result. We use Adam optimizer with learning rate = 0.001 and batch size = 64 across the experiments. We set $(\beta_1, \beta_2) = (0.5, 0.9)$ for ICM and set $(\beta_1, \beta_2) = (0.9, 0.999)$ for VAEs as Gulrajani et al. (2017) suggest. We train the VAEs with dimension $[10, 20, 30, 50]$. We find the VAEs with 20 dimensions have the best robustness when $\beta$ is optimal. For the ICM model, we set the dimension of each $z_{G_i}$ to 1, 2, and 1 on MNIST, FashionMNIST, and Sprites, respectively. The dimension of $z_T$ is 4, 5, and 3 on MNIST and FashionMNIST, and Sprites, respectively.

### B.3. Disentangled Representations Datasets

The disentangled representations datasets, such as 3DShapes, assume the latent explanatory factors are independent of each other. Such an assumption is not compatible with the class of system that we study. We tried to create a dataset that correlates one type of variation (e.g. floor hue) with one shape. However, such a configuration will make the data too small to use.

### B.4. Choosing the Number of Mechanisms

Our method does not require users to know the precise number of mechanisms in advance because our ICM model can tolerate the discrepancy between the learned mechanisms and the true mechanisms if the number of learned mechanisms is greater or equal to the number of true mechanisms. For example, we use 15 mechanisms in the MNIST experiment. The users can easily choose the number of mechanisms by applying the following procedure: 1) Pick a random number and train the ICM model. 2) Do a latent space traversal and observe the type of variations. 3) If the data samples from the same data generating mechanism share the same type of variations, the ICM model is ready to use. Otherwise, increase the number of mechanisms and repeat steps 1)&2).

### B.5. Training Setting for Sprites

If the environment shift that affects both the ICM model and the downstream GBT model is caused by color, we do not find disentanglement improves the accuracy of the GBT model. Dittadi et al. (2021) report similar results. The possible reason is that the color shift in our experiment is not continuous, differing from width and brightness shifts. If the ICM model has never seen any blue color, it is difficult to "imagine" a blue object based on the red or green. Therefore, we let the environment shift affect the GBT model only. Such a setting is also helpful for showing the benefit of disentanglement and is called "out-of-distribution 1 setting" in a recent work (Dittadi et al., 2021).

## Appendix C. More Experimental Results

### C.1. Ablation Study on Loss Terms

Figure 8 shows that removing the second term $\mathbb{E}_{z \sim p(z)} \| z_G - E(G(z))_G \|_2^2$ and the third term $\mathbb{E}_{z \sim p(z)} \| z_T - E(G(z))_T \|_2^2$ in the loss function results in entanglement between the data generating and transforming mechanisms. Specifically, the latent space traversal for the data transforming mechanism shows a variation that associate with the styles of digit 4. We note that if we remove

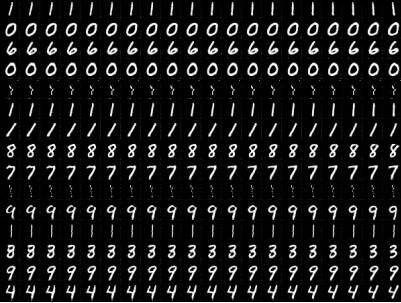

(a) Data Generating Mechanisms Subspaces, Each Row Represents One Dimension in $z_G$

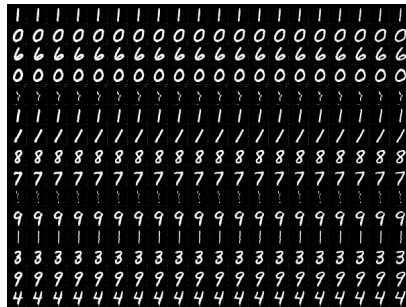

(b) Data Transforming Mechanism Subspace, Dimension #0, Entangled with Data Generating Mechanisms.

Figure 8: Latent Space Traversal of ICM model on MNIST. The ICM model is trained without the second term $\mathbb{E}_{z \sim p(z)} \|z_G - E(G(z))_G\|_2^2$ and the third term $\mathbb{E}_{z \sim p(z)} \|z_T - E(G(z))_T\|_2^2$ in the loss function.

the second and third terms and keep the fifth term $\mathbb{E}_{x \sim p(x)} \|x - G(E(x))\|_2^2$, the training of the generative model fails. One possible reason is that the encoder $E$ can not predict $z$ accurately without the second and third terms in the loss function, making $E(x)$ meaningless and misleading the training. Therefore, we have to remove the fifth term in the ablation study for the second and third terms. Figure 9 shows that removing the fourth term $\mathbb{E}_{z \sim P(z)} \mathcal{H}(z_C, E(G(z))_C)$ results in entanglements between data generating mechanisms. If we remove the fifth term $\mathbb{E}_{x \sim p(x)} \|x - G(E(x))\|_2^2$, Figure 10 shows that the generative model may miss some mechanisms (e.g., the mechanism for digit 5 in the attached figure).

## C.2. Ablation Study on Degenerate Mixture

The visualization results in Figure 11 suggest that if we remove $z_G$ from the prior, the generative model is not robust to interventions. We could observe entanglement between digits 3 and 8, 3 and 5, 4 and 9, 8 and 2. One possible reason is that the additional task of predicting $z_G$ benefits the disentanglement.

## C.3. Comparison between Degenerate and Non-degenerate Mixture Priors

We conduct qualitative ablation study using a non-degenerate mixture prior $p(z_G) = \sum_{i=1}^{K_G} \frac{1}{K_G} \cdot \mathcal{N}(z_G \mid \mu_i, \sigma_i)$ in the ICM model. We find that if we want to reduce the overlap between mixture components by letting $\mu_i$ be far away from each other and letting each $\sigma_i$ be small, the samples from clusters with large $|\mu_i|$ makes the GAN training unstable, and the small $\sigma_i$ reduces the model capacity for each mixture component. Therefore, we choose a smaller $K_G=3$, let $\mu = [-1.0, 0.0, 1.0]$ and $\sigma = [0.5, 0.5, 0.5]$, and report the visualization result in Figure 12(a). We can see that (1) the data variability is limited, (2) the two types of variations, circle size and circle number, are entangled, and (3) interventions could change the label of the digit. (2) and (3) support our discussion in Section 2.2.

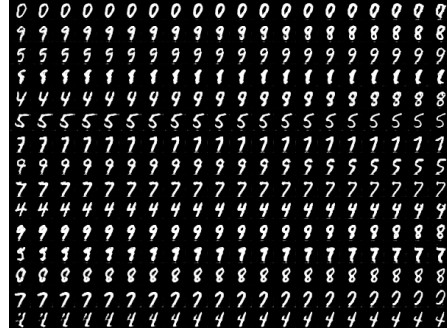

Figure 9: Latent Space Traversal of ICM model on MNIST, Data Generating Mechanisms Subspaces, Each Row Represents One Dimension in $z_G$. The ICM model is trained without the fourth term $\mathbb{E}_{z \sim P(z)} \mathcal{H}(z_C, E(G(z))_C)$ in the loss function.



Figure 10: Latent Space Traversal of ICM model on MNIST, Data Generating Mechanisms Subspaces, Each Row Represents One Dimension in $z_G$. The ICM model is trained without the fifth term $\mathbb{E}_{x \sim p(x)} ||x - G(E(x))||_2^2$ in the loss function.

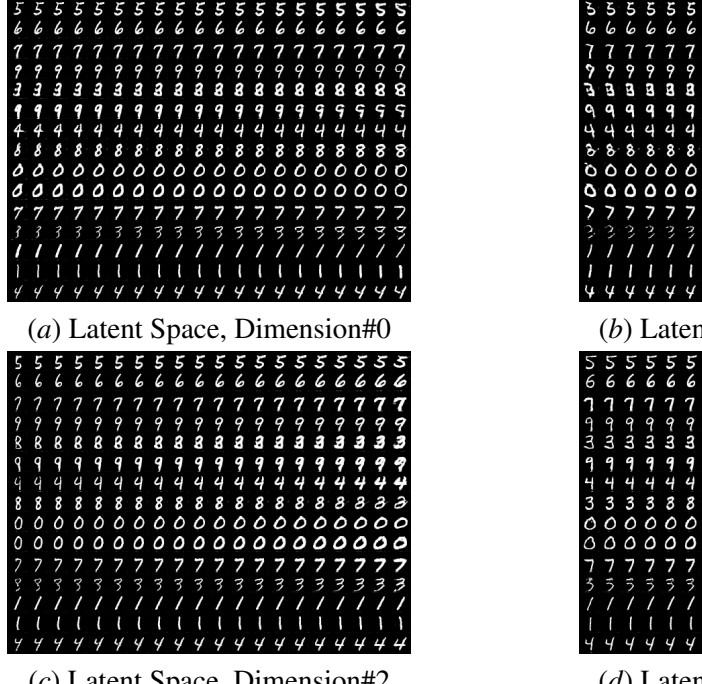

(a) Latent Space, Dimension#0

(b) Latent Space, Dimension#1

(c) Latent Space, Dimension#2

(d) Latent Space, Dimension#3

Figure 11: Latent Space Traversal of ICM Model on MNIST. The ICM model is trained without $z_G$.

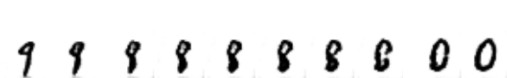

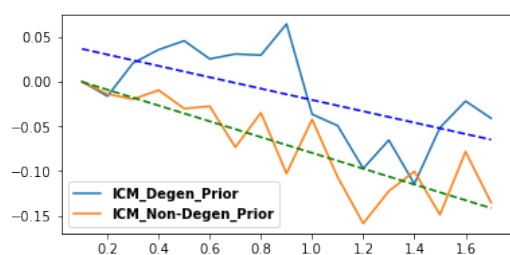

(*a*) Latent Space Traversal of ICM Model with Non-degenerate Mixture Prior.

(*b*) GBT Accuracy Decrease (y-axis) under Different Environment Shift Distance (x-axis).

Figure 12: (a) shows that circle size factor and circle number factor are entangled. (b) shows that ICM model with degenerate mixture prior slows down the accuracy decrease under environment shift.

Quantitatively, we consider a binary classification task using digits 4 and 9 from the MNIST dataset. The shifted conditions are the styles of digits 4 and 9 (e.g., the circle size of digit 9). For the train set, we collect $\{x \mid x \in \mathcal{X}, \hat{z}_4 = E(x)_4 \in [C_{lower}, C_{upper}) \vee \hat{z}_9 = E(x)_9 \in [C_{lower}, C_{upper})\}$, where $[C_{lower}, C_{upper})$ is $[-1.0, 1.0)$. We collect a union of two subsets for each test set by letting $[C_{lower}, C_{upper})$ be $[-1.1, -1.0), [-1.2, -1.1), ..., [-3.0, -2.9)$ and $[1.0, 1.1), [1.1, 1.2), ..., [2.9, 3.0)$, respectively. We use the GBT model as the classifier. Figure 12(*b*) shows that as the environment shifts, the accuracy decreases faster if we use the encoder from the ICM model with a non-degenerate mixture prior. Additionally, using degenerate mixture prior yields 90.72% average accuracy, contrasting the 78.44% average accuracy from using non-degenerate mixture prior. This result complements our discussion in Section 2.2.

## C.4. Qualitative Evaluation

**MNIST** We visualize the data transforming mechanism subspace traversal in Figure 15.

**FashionMNIST** We visualize the traversal for each data generating mechanism subspace and the data transforming mechanism subspace in Figure 16.

**Sprites** We visualize the traversal for each data generating mechanism subspace and the data transforming mechanism subspace in Figure 17.

## C.5. Quantitative Evaluation

We further investigate our model under MNIST (R) environment shift. Figure 13 shows the accuracy changes as the shift strength increases. Our method yields higher accuracy and slower accuracy decreases in both experiments when the shift strength is moderate. However, after a passing threshold, our method begins to lose its advantage. Our analysis suggests two reasons: 1) The test data shifts too far from the training data, and the base generative model can not generalize to the test sets. After the test set shifts too far away, none of the methods perform well. Thus, it is hard to conclude that a model with $\sim 40\%$ accuracy is better than a model with $\sim 35\%$ accuracy. 2) The

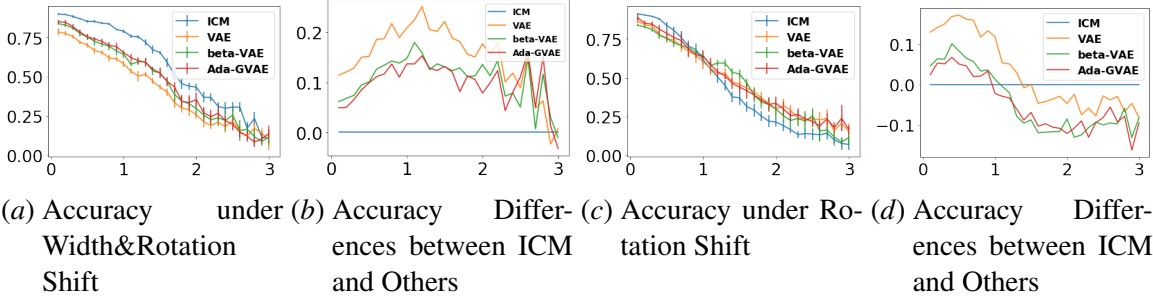

(a) Accuracy under Width&Rotation Shift    (b) Accuracy Differences between ICM and Others    (c) Accuracy under Rotation Shift    (d) Accuracy Differences between ICM and Others

Figure 13: Accuracy and accuracy differences of downstream classifiers under environment shift on the MNIST dataset. If ICM outperforms another method, the accuracy difference is positive. Note that our proposed ICM outperforms all the baselines in most settings. Exceptions include when the environment shifts too far for any model to tolerate.

test set contains too few samples, just around tens or hundreds, making the evaluation inaccurate. As we can see from Figure 13, the larger the shift, the bigger the accuracy variations.

To eliminate this interference, we measure how much environment shift is needed to decrease the accuracy by a relative percentage. Such evaluation will put more weight on the test set that has more samples and yields reasonable accuracy. Tables 4, 5, and 6 show our method can *tolerate more environment shift* before the test sets shift too far away and decrease the accuracy by 10%, 20%, and 40%, relatively.

Table 4: Shift Distance Needed for 10% Relative Accuracy Drop under Environment Shift

| MODEL | MNIST (T) | MNIST (W&R) | MNIST (R) | FASHIONMNIST (D&W) |
|---|---|---|---|---|
| ICM | **1.4** | **0.8** | **0.6** | **0.4** |
| VAE | 0.6 | 0.4 | 0.4 | 0.2 |
| $\beta$-VAE | 1.1 | 0.5 | 0.4 | 0.3 |
| ADA | 1.1 | 0.5 | 0.4 | 0.3 |

Table 5: Shift Distance Needed for 20% Relative Accuracy Drop under Environment Shift

| MODEL | MNIST (T) | MNIST (W&R) | MNIST (R) | FASHIONMNIST (D&W) |
|---|---|---|---|---|
| ICM | **2.0** | **1.2** | **0.8** | **0.7** |
| VAE | 1.2 | 0.7 | 0.7 | 0.4 |
| $\beta$-VAE | 1.7 | 0.8 | 0.7 | 0.5 |
| ADA-GVAE | 1.6 | 0.9 | 0.7 | 0.5 |

Table 6: Shift Distance Needed for 40% Relative Accuracy Drop under Environment Shift

| MODEL | MNIST (T) | MNIST (W&R) | MNIST (R) | FASHIONMNIST (D&W) |
|---|---|---|---|---|
| ICM | **2.6** | **1.6** | 1.1 | **1.2** |
| VAE | 1.9 | 1.3 | 1.1 | 0.8 |
| $\beta$-VAE | 2.4 | 1.5 | **1.3** | 1.1 |
| ADA-GVAE | 2.2 | 1.5 | 1.2 | 1.0 |

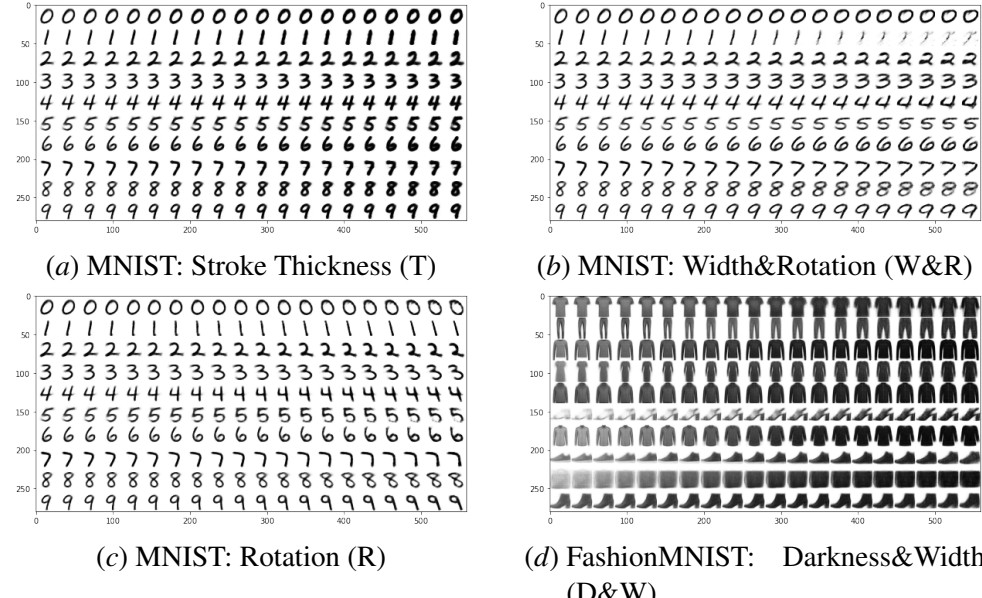

(a) MNIST: Stroke Thickness (T)  (b) MNIST: Width&Rotation (W&R)

(c) MNIST: Rotation (R)  (d) FashionMNIST: Darkness&Width (D&W)

Figure 14: Conditions (Latent Variables) Used in the Environment Shift Experiment.

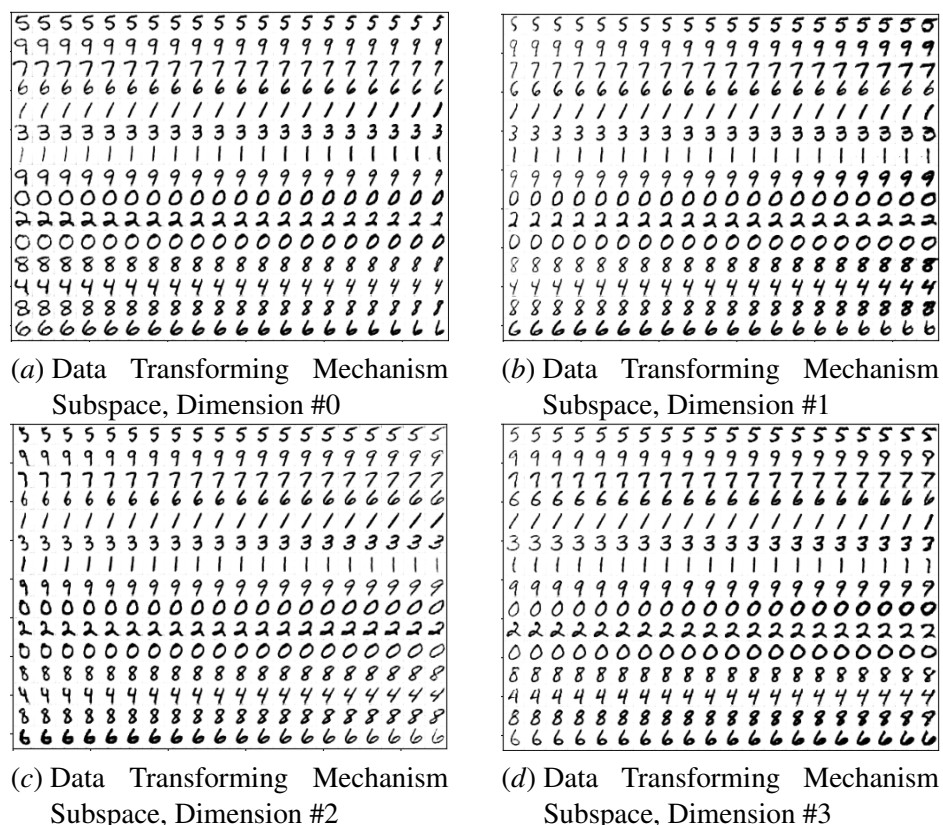

(*a*) Data Transforming Mechanism Subspace, Dimension #0

(*b*) Data Transforming Mechanism Subspace, Dimension #1

(*c*) Data Transforming Mechanism Subspace, Dimension #2

(*d*) Data Transforming Mechanism Subspace, Dimension #3

Figure 15: Data Transforming Mechanism Subspace Traversal of ICM model on MNIST

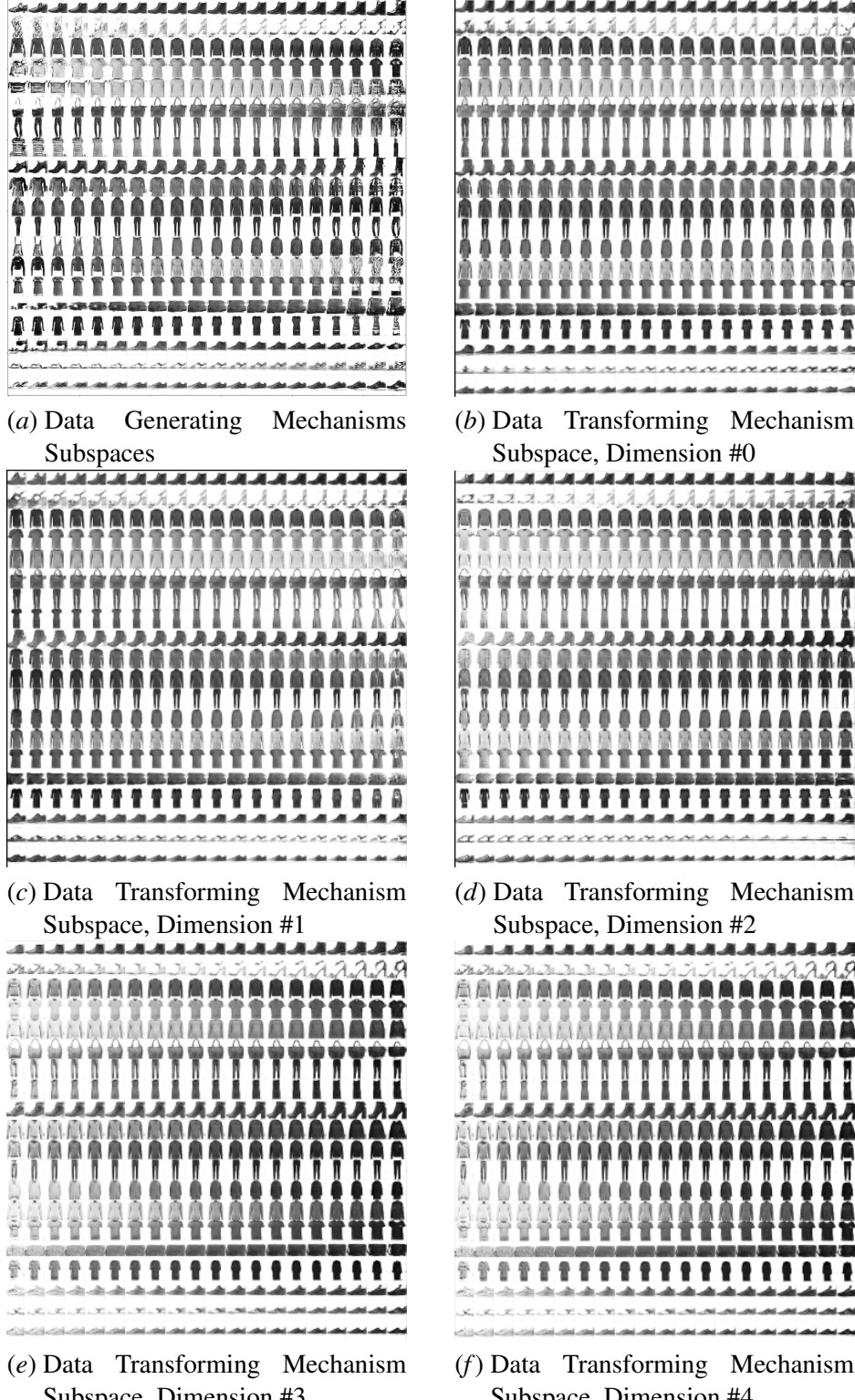

(*a*) Data Generating Mechanisms Subspaces

(*b*) Data Transforming Mechanism Subspace, Dimension #0

(*c*) Data Transforming Mechanism Subspace, Dimension #1

(*d*) Data Transforming Mechanism Subspace, Dimension #2

(*e*) Data Transforming Mechanism Subspace, Dimension #3

(*f*) Data Transforming Mechanism Subspace, Dimension #4

Figure 16: Latent Space Traversal of ICM model on FashionMNIST

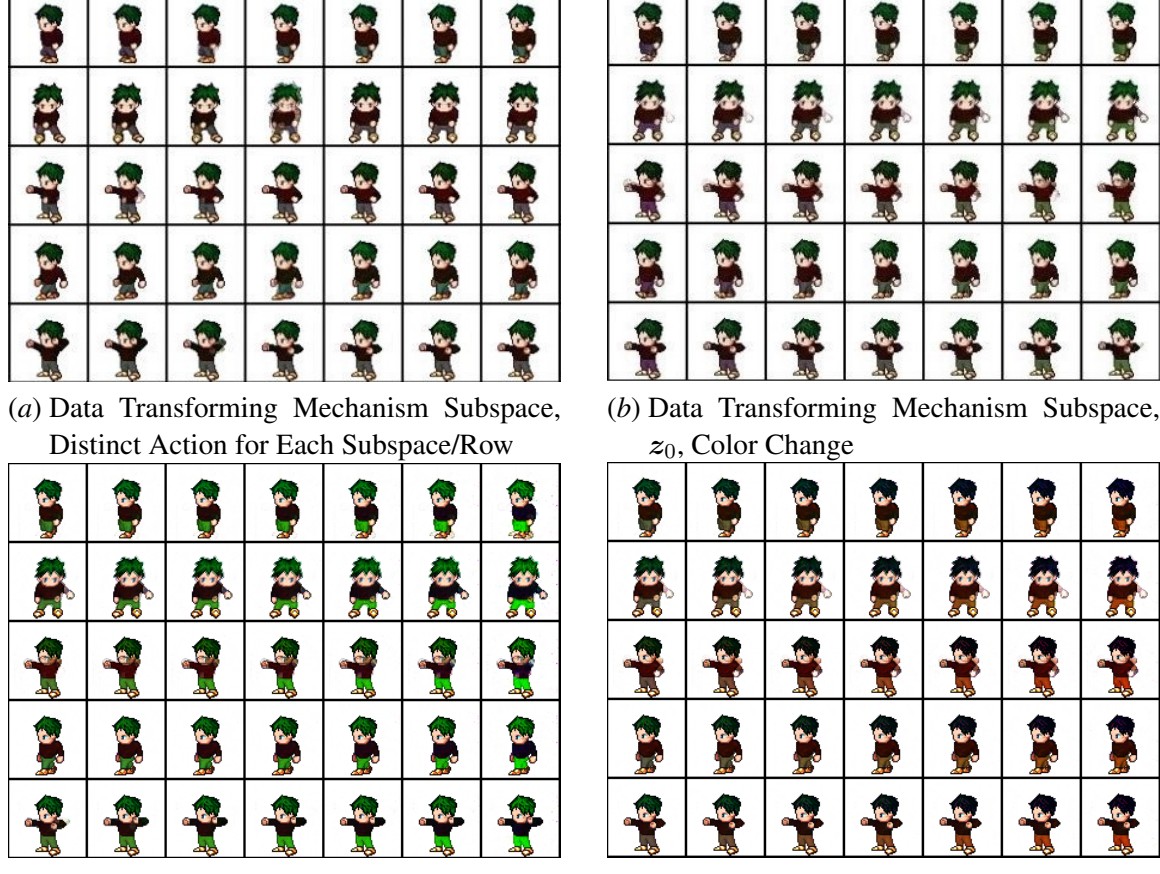

(*a*) Data Transforming Mechanism Subspace, Distinct Action for Each Subspace/Row

(*b*) Data Transforming Mechanism Subspace, $z_0$, Color Change

(*c*) Data Transforming Mechanism Subspace, $z_1$, Color Change

(*d*) Data Transforming Mechanism Subspace, $z_2$, Color Change

Figure 17: Latent Space Traversal of ICM model on Sprites

