# OpenReview forum: "Identifying Coarse-grained Independent Causal Mechanisms with Self-supervision"
_cclear.cc/CLeaR/2022/Conference — CLeaR 2022 Poster_

### Official Review · Reviewer_6Xbo · 2021-11-21

**Confidence:** 4
**Overall Score:** 5

**Main Review:**

First, I will start with some positive sides of the paper, especially regarding its originality and significance, then later in technical quality & correctness, I will point out some serious concerns I have about the proofs, that I hope the authors can clarify for me.

### Originality

Not being very familiar with the literature from disentangled representation learning, I cannot fully attest to the originality of the proposed work. However, the paper seems to clearly explain and outline their contribution compared to previous works. Their proposed prior is quite novel and their theoretical **claims** seem to advance the state of the art concerning identifiability issues in disentangled representation learning (I will come back on this later). Finally, the proposed self-supervised loss is an original way of restricting the generative model so that it falls in line with the assumptions of the identifiability proof. All in all, the paper is far from lacking original material.

### Significance

The paper addresses the problem of disentangled representation learning, where the aim is to learn a representation with independent subcomponents. This is an important problem since it would allow a much better interpretation of the learned representation along with more robust downstream models. There are tight links with causal learning and independent causal mechanisms, that are discussed in the paper and related works, making this topic very relevant to the CLeaR community.

### Technical quality & Correctness

I have tried to step through the proofs and have several concerns :

- In the proof of the lemma, I don’t see how h can be anything else than ${g^*}^{-1}\circ g$ ? Since both $g$ and $g^*$ are invertible and have the same domain, there exists one and only one function transforming one into the other. In which case the whole proof can be reduced to that. Again, I might be misunderstanding something here.
- It is not clear to me if the domain of the true generative model (and the learned one) is exactly the support of the degenerate mixture prior, that I will call $S$ (which is the set that has a structure constraint imposed) or on all $Z$ (which is defined in the paper as the cross product of $Z_{G_1}, Z_{G_2}, …$ etc). The cross product clearly doesn’t have the structure constraint.
    - If the former is the case, $Z$ should just be defined as $S$ instead of the cross product. This confusion raises many questions for the rest of the proof.
    - For example for $h$: is it a permutation of the cross product set $Z$ or $S$?
        - If $h$ goes from $Z$ to $Z$, the image of $S$ under $h$ has no reason of being structure constrained. For instance, in a bi-dimensional case with 2 generative mechanisms, and ignoring the transformative component, one could choose $h$ to be the invertible linear transformation represented by the matrix $\pmatrix{1 & 1 \\\\ 0 & 1}$ which would give $h(x)=\pmatrix{x_{G_1} + x_{G_2} \\\\ x_{G_2}}$. This clearly doesn’t satisfy the structure constraint.
        - If $h$ goes from $S$ to $S$ and is invertible, then $h(S)=S$ so the image of $S$ trivially satisfies the structure constraint.
        - Although you define $\hat Z$ as being the image of $Z$ under $h$, the notation $\hat z_{g_i}$ is ill-defined if $Z$ is the cross product since the image of $Z_{G_i}$ has no reason of being on a given set of coordinates in the new space. (See counter-example above).
- All these contradictions lead me to believe that the authors actually meant that $g$ goes from $S$ to $X$ and that $h$ goes from $S$ to $S$. In this case, I fully agree that $h$ won’t entangle different generative mechanisms if it is a smooth invertible function from $S$ to $S$, even if the proof should be clarified a little. However, for the transformative part, if we consider a bi-dimensional case with the first dimension being the only generative one and the second dimension to be the transformative one, S would be the cross product of $Z_{G_1}$ and $T$. And if we take again $h$ to be the linear transformation $\pmatrix{1 & 1 \\\\ 0 & 1}$, then $h$ clearly entangles the generative and transformative input?
    - So although the motivation for separating transforming mechanism and generating mechanism is well explained and intuitively makes sense, it seems there isn’t independence in the variability they express. For instance, in the latent space traversals in Figure 6, we can see that changing the generative latent variables changes the style of the number but also the thickness and rotation (which are supposed to be encoded by the transformative part).
    - Other than the entanglement of the generative and transformative parts within each generative component, the model seems to work really well in the disentanglement between the different generative mechanisms and the traversals of the latent space are quite impressive! Maybe a crucial component of this setup is the one-hot vector appended to the latent variables. Enforcing the invertibility of the generative model with the one-hot vector present in the input might be the thing forcing the model to not entangle different generative mechanisms. I think it would be important to have an ablation with this one-hot vector but without the degenerate mixture prior, and the other way around, to see which of the two selects the independent generative mechanism. If the model works without the degenerate mixture and with the presence of the one-hot vector (traversals of the transformative part don’t generate different digits for example), this would remove the need for the degenerate mixture, hence simplifying the model.
    - I don’t think this is the first time one-hot vectors were used to condition the generative model but imposing the invertibility of the generative model through their self-supervised loss is a very original approach and I really believe the authors have come up with something very interesting that warrants further investigation!
- There are two scenarios here
    - If the one-hot vector proves to be the key to the disentanglement, then getting rid of the degenerate mixture prior and the generative latent variables won’t do any harm, and the “transformative” latent variables would be the only ones expressing the variability of the generative components, while the one-hot encoding vector would be the one choosing which component is currently active. The proofs for identifiability would need to be rewritten by taking into account the one-hot vector instead of the degenerate mixture model.
    - If not, then getting rid of the transformative latent variables would make the current version of the proof more correct and would indeed make the true generative model identifiable.

But in any case, in the way it is presented currently, I don’t think the paper is theoretically sound, the notation in the proofs isn’t very clear, and the reasoning is not very rigorous. I am really hoping that the authors have further clarifications that would render my concerns null since I really admire the empirical results and I think that some of the ideas behind the paper, even without the theoretical part, are very interesting and merit to be published. I would be very happy to update my score should the rebuttal prove me wrong.

**Summary:**

In a classification setting, it is often supposed that the data comes from a mixture distribution with separated components. This paper proposes to model each component as the result of a composition of independent generative and transformative models. The generative models take as input low dimensional latent variables that are disentangled from one another. This data-generating process is inspired by the Independence of Causal Mechanisms. Often in generative modeling, latent variables are drawn from a prior distribution and are fed as input to the generative model. When trying to learn disentangled latent variables, a common practice is to use a disentangled prior that makes the latent variables independent from one another. This paper proposes a novel “degenerate mixture” prior that only generates a single latent component and sets the rest to 0. With this degenerate mixture prior, the authors prove that, under some assumptions, the true generative model is identifiable up to a smooth invertible transformation of the latent space. The main assumption for the identifiability result is that the generative model is a diffeomorphism (smooth invertible and with smooth inverse). Therefore, they propose a self-supervised loss to enforce this assumption on a learned generative model. They show qualitatively that the latent variables are indeed disentangled by traversing each subspace and showing that the generated samples belong to a single class. Finally, they quantitatively evaluate their method by studying the robustness of a classifier trained on their learned representation under environment shift.

---

> ### Author Response · Authors · 2021-12-03
> **Response to Reviewer 6Xbo**
>
> We thank the reviewer for the constructive feedback! We answer the concerns from the reviewer in the following:
>
> &nbsp;
>
> **Comment:**
>
> The proof of the lemma.
>
> **Response:**
>
> The reviewer’s understanding of Lemma 3 is correct.
>
> &nbsp;
>
> **Comment:**
>
> It is not clear to me if the domain of the true generative model (and the learned one) is exactly the support of the degenerate mixture prior, or all on $\mathcal{Z}$.
>
> **Response:**
>
> Section 3.2 and Theorem 4 operate under the definition of the degenerate mixture prior in Section 2.2. We will add the definition in Section 2.2 to Theorem 4. Since the inputs of function $g^*$ are samples from the degenerate mixture prior, the domain $\mathcal{Z}$ is the same as the support of the latent variable $\mathbf{z}$, which implies that $\mathcal{Z}$ is structurally constrained. Since both the true generative model $g^*$ and the estimated generative model $g$ take the same latent variable $\mathbf{z}$ as input, we have $\mathcal{Z} = \hat{\mathcal{Z}}$, which implies that $\hat{\mathcal{Z}}$ is also structurally constrained.
>
> &nbsp;
>
> **Comment:**
>
> However, for the transformative part, if we consider a bi-dimensional case with the first dimension being the only generative one and the second dimension to be the transformative one, $S$ would be the cross-product of  $Z_{G_i}$ and $T$. And if we take again $h$ to be the linear transformation $\begin{pmatrix} 1 & 1 \\\ 0 & 1 \end{pmatrix}$, then $h$ clearly entangles the generative and transformative input?
>
> **Response:**
>
> The latent variable $\mathbf{z}$ does not follow a mixture distribution in the counter-example that the reviewer raises. Therefore, the counter-example is not valid. If we assume the latent variable $\mathbf{z}$ follows a degenerate mixture distribution that has two mixture components and let $h(\mathbf{z}) = \begin{pmatrix} 1 & 0 & 0 \\\ 0 & 1 & 1 \\\ 0 & 0 & 1 \end{pmatrix} \cdot \mathbf{z}$, then if we let $\mathbf{z} = [1, 0, 1]^\top$, we have $\hat{\mathbf{z}} = h(\mathbf{z}) = [1, 1, 1]^\top$, violating the structural constraint.
>
> &nbsp;
>
> **Comment:**
>
> In the latent space traversals in Figure 6, we can see that changing the generative latent variables changes the style of the number but also the thickness and rotation (which are supposed to be encoded by the transformative part).
>
> **Response:**
>
> The latent space traversal is done with an estimated generative model, which may not converge to the global minimum and show minor entanglement. We will investigate the optimization issue further in the future. We note that our visualization results in Figure 6 are still SOTA regarding disentangling data generating and data transforming mechanisms.
>
> &nbsp;
>
> **Comment:**
>
> I think it would be important to have an ablation with this one-hot vector but without the degenerate mixture prior, and the other way around, to see which of the two selects the independent generative mechanism.
>
> **Response:**
>
> We attached our ablation results via an anonymous link: https://drive.google.com/file/d/1LwoAZYpN8ANlZy5--Q-hXjgts6flINO5/view?usp=sharing. The visualization result suggests that an ICM model with the one-hot vector but without the degenerate mixture prior is not robust to interventions, and the additional task of predicting the latent variables $\mathbf{z}_G$ may benefit the disentanglement.
>
> &nbsp;
>
> **Comment:**
>
> There are two scenarios here.
>
> **Response:**
>
> We thank the reviewer for making various suggestions! However, since the additional ablation study suggests that getting rid of the degenerate mixture prior is harmful, and our proof for the transformation mechanisms still holds, we decide to keep our approach and theoretical analysis. We would be happy to address any further concerns from the reviewer.

---

> > ### Comment · Reviewer_6Xbo · 2021-12-15
> > **Correction to counter example and other remarks**
> >
> > Thank you for your reponse,
> >
> > Indeed the counter example I gave doesn't work, however tweaking just a little seems to reintroduce the problem. If we let $h=\pmatrix{1&0&0 \\\\ 0&1&0 \\\\ 1&0&1}$ then $h(z)=\pmatrix{z_{G_1} \\\\ z_{G_2} \\\\ z_{G_1}+z_{T}}$ which clearly entangles $Z_{G_1}$ and $Z_T$ while still abiding to the structure constraint if I'm not mistaken.
> >
> > For the ablation, do you confirm you only remove $z_G$ and keep the one hot encoding and everything else? I ask because the dimensionality of $z_T$ in your ablation is 5 while it is 4 for the paper. So I would like to see the ablation where you *only* remove $z_G$.
> >
> > Also, I am now unclear on how the latent space traversal is done exactly. There are as many rows as there are mechanisms so I am guessing that you set the one hot encoding to the correct mechanism for each row? When doing a traversal of $Z_T$, what value do you give $z_G$ for each row? Conversely, when doing a traversal on $Z_G$, what value do you give $z_T$?
> >
> > Again, I genuinely appreciate this work, am impressed by the results and would love to see it go further. Is a release of the code planned at some point?
> >
> > I would like to point out that my below acceptance threshold rating is mainly due to the lack of rigor in the proofs and confusing notation, Reviewer1 seems to agree on this point.

---

> > > ### Author Response · Authors · 2021-12-18
> > > **Response to Reviewer 6Xbo**
> > >
> > > Thank you for your additional constructive remarks on our paper!
> > >
> > > &nbsp;
> > >
> > > **Comment:**
> > >
> > > Indeed the counter example I gave doesn't work, however tweaking just a little seems to reintroduce the problem.
> > >
> > > **Response:**
> > >
> > > Thank you for raising the counter example. We have fixed a typo in our theorem regarding the function $h$.
> > > 1. In Theorem 4, we now state “$h$ maps each $\mathcal{Z} _{G_i}$ to $\hat{\mathcal{Z}} _{G_i} \times \hat{\mathcal{Z}}_T$, disjoint from $\hat{\mathcal{Z}} _{G_j}, \forall j \neq i$ and maps $\mathcal{Z} _T$ to $\hat{\mathcal{Z}} _T$, disjoint from $\hat{\mathcal{Z}} _G$”. Here, the $\mathcal{Z} _{G_i}$ is disentangled from $\mathcal{Z} _ {G_j}, \forall j \neq i$ and $\mathcal{Z} _T$ in $\hat{\mathcal{Z}} _{G_i}$. This result is consistent with our goal of disentangling $\mathcal{Z} _T$ and $\mathcal{Z} _{G_j}, \forall j \neq i$ from $\mathcal{Z} _{G_i}$.
> > > 2. As consequences, we will (1) update the introduction correspondingly, (2) mention that $h$ could map $\mathcal{Z} _{G_i}$ to $\hat{\mathcal{Z}} _T$ and $\hat{\mathcal{Z}} _{G_i}$ in the proof, and (3) add the following discussion to our experiment: “Since the true $\mathbf{z} _T$ is uniquely determined by $\hat{\mathbf{z}} _T$ and the true $\mathbf{z} _{G_i}$, therefore, for any data sample $\mathbf{x} = g(h(\mathbf{z}))$ with a given $\mathbf{z} _{G_i}$, varying $\hat{\mathbf{z}} _T$ is equivalent to varying the true $\mathbf{z} _T$”.
> > >
> > > &nbsp;
> > >
> > > **Comment:**
> > >
> > > For the ablation, do you confirm you only remove $z_G$ and keep the one hot encoding and everything else?
> > >
> > > **Response:**
> > >
> > > 1. In the previous ablation study, we increase the dimension of $\mathbf{z}_T$ by 1 as compensation for removing $\mathbf{z}_G$ and keep the number of input latent variables for generating an observational data sample unchanged.
> > >
> > > 2. We provide additional ablation results (https://drive.google.com/file/d/1NMik5e38zAwUPvDPxgd6NBIf5yshvqGp/view?usp=sharing) under the condition that the reviewer suggests, showing that removing $\mathbf{z}_G$ hurts the disentanglement.
> > >
> > >
> > > &nbsp;
> > >
> > >
> > > **Comment:**
> > >
> > > Also, I am now unclear on how the latent space traversal is done exactly.
> > >
> > > **Response:**
> > >
> > > 1. We set the one hot encoding to the correct mechanism for each row.
> > > 2. When doing a traversal of $\mathbf{z}_T$, we set $\mathbf{z} _G$ to $\mathbf{0}$.
> > > 3. When doing a traversal of $\mathbf{z}_{G_i}$, we set $\mathbf{z} _T$ and $\mathbf{z} _{G_j}, \forall j \neq i$ to $\mathbf{0}$.
> > >
> > > &nbsp;
> > >
> > > **Comment:**
> > >
> > > Again, I genuinely appreciate this work, am impressed by the results and would love to see it go further. Is a release of the code planned at some point?
> > >
> > > **Response:**
> > >
> > > Thank you for your encouragement! Yes, we plan to release the code upon publication.
> > >
> > > &nbsp;
> > >
> > > **Comment:**
> > >
> > > I would like to point out that my below acceptance threshold rating is mainly due to the lack of rigor in the proofs and confusing notation, Reviewer1 seems to agree on this point.
> > >
> > > **Response:**
> > >
> > > It is clear the reviewer agrees that this work is valuable and timely. We hope the reviewer agrees that the rigor and notation issues are easy fixes that have been addressed in the rebuttal, and the work is worthy of publication.

---

### Official Review · Reviewer_Na2R · 2021-11-22

**Confidence:** 4
**Overall Score:** 9

**Main Review:**

This is a hard problem, and the algorithm in this paper achieves success by leveraging the assumption of independent generating mechanisms. Theoretical characterizations, a simulation study, and performance on three standard benchmarks are all provided.

Overall, I felt that this was a very strong paper. It is clearly written (or as clearly as possible given the complexity of the algorithm), makes progress on a significant problem, and provides some useful tests. It would have been nice to have more systematic simulation testing, not just on a case that is particularly favorable to this method (rather than standard VAE). I was also surprised that there was no reference to recent work on clustering datapoints based on shared causal structure (e.g., Huang, et al’s 2019 NeurIPS paper on shared causal relation clustering). Obviously, that challenge is slightly different from this one, but there are many similarities and overlaps, so it would be helpful to have some connections with that type of research.

**Summary:**

This paper develops a method for discovering latent generating mechanisms, perhaps with post-discovery distribution shifts, from an unlabeled mixture dataset.

---

> ### Author Response · Authors · 2021-12-03
> **Response to Reviewer Na2R**
>
> We thank the reviewer for the positive review! We are glad the reviewer found the paper solid and relevant.
>
> &nbsp;
>
> **Comment:**
>
> It would have been nice to have more systematic simulation testing, not just on a case that is particularly favorable to this method (rather than standard VAE).
>
> **Response:**
>
> We compared our method to Ada-GVAE, a SOTA identifiable VAE. We also included results from prior work on learning data generating mechanisms in Figure 1 as comparisons.
>
> &nbsp;
>
> **Comment:**
>
> I was also surprised that there was no reference to recent work on clustering datapoints based on shared causal structure (e.g., Huang, et al’s 2019 NeurIPS paper on shared causal relation clustering).
>
> **Response:**
>
> We thank the reviewer for pointing out the closely related paper. We will add the paper to the related work section.

---

### Official Review · Reviewer_1JUg · 2021-11-23

**Confidence:** 4
**Overall Score:** 5

**Main Review:**

The approach this paper suggests is interesting, but I have two main criticisms that prevent me from recommending acceptance. The first one being the overall lack of clarity and technical rigor of the paper. The second point is that this algorithm would be best described as a clustering algorithm than a disentanglement algorithm, and should be acknowledged as such both in its presentation and in its empirical evaluation by comparing to SOTA clustering algorithms. See below for details.

### Originality:
The approach is interesting and, to the best of my knowledge, novel.

### Significance:
The degenerate priors idea is simple and seems to yield good qualitative results on simple tasks like MNIST, FashionMNIST and Sprites. That being said, the baselines were not well tailored to the tasks at hand. I believe state-of-art methods in clustering would make stronger baselines, like VaDe from "Variational Deep Embedding: An Unsupervised and Generative Approach to Clustering" by Jiang et al.(2017). Significance would be easier to assess with a more suited set of baselines to compare the authors' algorithm to.

### Clarity:
- Section 2.1: It seems like a lot of unnecessary notation is introduced that adds almost nothing to the understanding of the algorithm, for example the submodules $M\_{G\_i}$'s and $M\_{T\_i}$'s. Figure 2 added to my confusion. The actual model proposed is explained in Section 2.2, in a much simpler way, using the formulation with the mixture of degenerate gaussians as prior. I would get to this explanation much earlier in the text, to avoid requiring unnecessary work from your reader.
- The introduction could also be shortened, since a lot of what is discussed is related to the model that the reader has not encountered yet.

### Technical quality:
- I believe the presentation of the algorithm does not requires the disentanglement machinery and would be more easily explained as a clustering approach: each subspace of Z corresponds to a cluster, e.g. a digit in MNIST.
- MNIST and FashionMNIST do not lend themselves very well to disentanglement and are better modelled by clustering approaches. This is why I am not surprised that the ICM method performs better than the VAE and $\beta$-VAE which are disentanglement methods. I would like to see the proposed approach being compared to state-of-the-art clustering methods.
- Definition 1 is vacuous in the sense that any pair of diffeomorphisms (i.e. differentiable bijections with a differentiable inverse) will be equivalent. A similar point can be made about Definition 2: it is a very weak notion of identifiability. Weirdly, these definitions are not referenced in a later result, which makes them superfluous.
- The statement of Theorem 4 is imprecise. For instance, it says "$h$ maps each $\mathcal{Z}\_{G\_i}$ to $\hat{\mathcal{Z}}\_{G\_i}$, disjoint from $\hat{\mathcal{Z}}\_{G_i}$, $\forall j \not= i$, ...". A first problem is that, since $h$ is a map from $\mathcal{Z}$ to $\mathcal{Z}$, it is unclear what is meant by "$h$ maps each $\mathcal{Z}\_{G\_i}$ to...". I think what is meant is that "$h$ maps $\\{0\\} \times ... \times \\{0\\} \times \mathcal{Z}\_{G_i} \times \\{0\\} \times ... \times \\{0\\}$ to ..." or something along these lines. A second problem is that the $\hat{\mathcal{Z}}\_{G_i}$ where not defined anywhere and, more importantly, the statement does not allow for an indeterminacy due to the ordering of the subspaces that cannot be identified. The statement should have been "there exists a permutation $\sigma$ such that $h$ maps each $\mathcal{Z}\_{G\_i}$ to $\hat{\mathcal{Z}}\_{G\_{\sigma(i)}}$", or something similar.
- Again in Theorem 4, in "if there exists a smooth invertible function $h$ such that $g = g^* \circ h$, then ...", the if-clause is always satisfied, since one can always choose $h = (g^*)^{-1} \circ g$. But of course, the then-clause can't hold for all pair of $(g, g^*)$. What is missing is a "Suppose $g(Z)$ has the same distribution as $g^*(Z)$, where $Z$ follows the degenerate prior defined in ...". The proof presents a similar lack of rigor.
- The loss in equation (4) has a huge number of terms. An ablation study showing the importance of all of them would be more convincing. I am not sure why the second and third terms are important for invertibility, the reconstruction loss (fifth term) should encourage invertibility, right?

# Response to rebuttal
I thank the authors for their response. At the time I wrote the review, the fact that the method disentangles cluster identity from "data transformation" was not clear to me. This is indeed a clear distinction from other clustering algorithms. I also appreciate the additional ablation study. Consequently I have decided to update my score from 4 to 5. I am not increasing my score further because the theory remains imprecise. For instance, the author's response did not address my point about definition 1 and 2 being completely vacuous. Also, the fact that no permutation appears in theorem 4 is worrisome and cast doubts on the validity of its proof (worries also shared by reviewer 6Xbo). I believe the approach suggested is promising and interesting, but its theoretical foundations has to be clarified before being published.

**Summary:**

This work introduces a novel generative model based on a prior specified as a mixture of degenerate gaussians and provide a proof of its identifiability. The authors proposed a training procedure that minimizes a regularized Wasserstein distance. The disentanglement property of their approach is qualitatively confirmed with latent traversals. The authors also argue that their disentangle representation is more robust to some class of dataset shifts.

---

> ### Author Response · Authors · 2021-12-03
> **Response to Reviewer 1JUg (2/2)**
>
> **Comment:**
>
> The loss in equation (4) has a huge number of terms. An ablation study showing the importance of all of them would be more convincing.
>
> **Response:**
>
> We attached the ablation result via an anonymous link: https://drive.google.com/file/d/13KnG5EdxdRhm4YA5p2lJU7BwHC4hAehP/view?usp=sharing, showing that all the terms are necessary, as is discussed in Section 4. Specifically, removing the second and third terms results in entanglement between the data generating and transforming mechanisms. Removing the fourth term results in entanglements between data generating mechanisms. If we remove the fifth term, the generative model may miss some mechanisms (e.g., the mechanism of digit 5 in the attached figure).
>
> &nbsp;
>
> **Comment:**
>
> I am not sure why the second and third terms are important for invertibility, the reconstruction loss (fifth term) should encourage invertibility, right?
>
> **Response:**
> 1. The second and third terms enforce the sampled $\mathbf{z}$ to be predictable from the generated data $\mathbf{x}$. Then, we could go from $\mathbf{x}$ to the sampled $\mathbf{z}$, which implies that the function $g: \mathcal{Z} \rightarrow \mathcal{X}$ is invertible.
> 2. The issue with the reconstruction loss (fifth term) is that it does not include any sampled $\mathbf{z}$. Therefore, it is possible to encode the data $\mathbf{x}$ to $\mathbf{z}'$, differing from the sampled $\mathbf{z}$, while succeeding in reconstruction but violating invertibility. The role of the reconstruction term is discussed in Section 5.

---

> ### Author Response · Authors · 2021-12-03
> **Response to Reviewer 1JUg (1/2)**
>
> We thank the reviewer for the constructive feedback. Below are our responses to the reviewer's comments:
>
> &nbsp;
>
> **Comment:**
>
> It seems like a lot of unnecessary notation is introduced that adds almost nothing to the understanding of the algorithm, for example the submodules $M_{G_i}$'s and  $M_{T_i}$'s.
>
> **Response:**
>
> The reviewer may have confused the assumed data generating process, defined in Section 2.1, and our algorithm. The $M_{G_i}$'s and  $M_{T_i}$'s are modules (i.e., the independent causal mechanisms) in the data generating process, which we aim to recover using our algorithm.
>
> &nbsp;
>
> **Comment:**
>
> The introduction could also be shortened, since a lot of what is discussed is related to the model that the reader has not encountered yet.
>
> **Response:**
>
> We will shorten the introduction as the reviewer suggests.
>
> &nbsp;
>
> **Comment:**
>
> I believe the presentation of the algorithm does not requires the disentanglement machinery and would be more easily explained as a clustering approach: each subspace of Z corresponds to a cluster, e.g. a digit in MNIST.
>
> MNIST and FashionMNIST do not lend themselves very well to disentanglement and are better modelled by clustering approaches. This is why I am not surprised that the ICM method performs better than the VAE and beta-VAE which are disentanglement methods. I would like to see the proposed approach being compared to state-of-the-art clustering methods.
>
> **Response:**
>
> Our approach differs from the clustering approaches as we propose to disentangle the data-transforming mechanisms from the data-generating mechanisms. The clustering approaches may recover the data-generating mechanisms but do not include any data-transforming mechanism. Specifically, in addition to the subspace for each cluster, we allocate a shared subspace, which is missing in prior clustering approaches, representing the data transforming mechanisms. The shared subspace encodes the data transformations related to the environment, such as lighting conditions and rotations. Disentangling the shared subspace from the subspaces for each cluster improves the robustness of the downstream model under environment shift, as is discussed in Section 1 and empirically verified in Section 6.3.
>
> As the reviewer suggests, we conduct additional experiments on VaDE, finding that VaDE does not outperform our method under environment shift, with the evaluation protocol in Section 6.3, as Table 1 shows. The reason is that VaDE does not disentangle the data transforming mechanisms, which are affected by the environment shift, from the data generating mechanisms, which are not affected by the environment shift.
>
> **&nbsp;Table 1**
>
> | MODEL| MN (T)&emsp;&emsp;&emsp;&emsp;|MN (W&R)&emsp;&emsp;&ensp;|MN (R)&emsp;&emsp;&emsp;&emsp;|FMN (D&W)&emsp;&ensp;|Sprites&emsp;&emsp;&emsp;&emsp;|Average|
>
> | ICM&emsp;&nbsp;&nbsp;| 74.63% +-2.34%| 56.91% +-3.70%|43.71% +-7.10%|46.61% +-5.24%|99.25% +-1.79%|64.22%&ensp;|
>
> | VaDE&emsp;|73.64%&nbsp; +-2.69%|48.57% +- 4.92%|48.54% +-7.16%|43.21% +-7.35%|98.25% +-2.95%|62.44%&ensp;|
>
> &nbsp;
>
> **Comment:**
>
> Definitions 1 and 2 are not referenced in a later result, which makes them superfluous.
>
> **Response:**
>
> The equivalence relationship defined in Definitions 1 and 2 is necessary for identifiability analysis. Our main identifiability result in Section 3.2 refers to Definitions 1 and 2. Lemma 3 first outlines conditions for $h$-identifiable, and Theorem 4 further shows that the function $h$ does not entangle the mechanisms.
>
> &nbsp;
>
> **Comment:**
>
> The statement of Theorem 4 is imprecise.
>
> **Response:**
>
> Thank you for your comment. Responses and planned additional clarification are as follows.
>
> 1. We believe “$h$ maps each $\mathcal{Z}_ {G_i}$ to $\hat{\mathcal{Z}}_{G_i}$” is the same as the reviewer’s suggestion.
> 2. $\hat{\mathcal{Z}}$ is the same as $\mathcal{Z}$.
> 3. To improve clarity, we will add a permutation function $\sigma$ as the reviewer suggests.
> 4. Section 3.2 and Theorem 4 operate under the definition of the degenerate mixture prior in Section 2.2. We will add the definition in Section 2.2 to Theorem 4. Since both the true generative model $g^*$ and the estimated generative model $g$ take the same latent variable $\mathbf{z}$ as input, $\mathbf{z}$ and $h(\mathbf{z})$ follow the same distribution. Therefore, $g^*(\mathbf{z})$ and $g(\mathbf{z}) = g^*(h(\mathbf{z}))$ follow the same distribution.

---

### Decision · Program_Chairs · 2022-01-12

**Decision:**

Accept (Poster)

**Comment:**

All reviewers have manifested that the empirical results in this paper are strong and worth discussing in the conference. While I suggest accepting (poster), I encourage the authors to take into account all the feedback given by the reviewers, since these have done a particularly good job in providing detailed comments.